# 4/4 and more, rhythmic complexity more strongly predicts groove in common meters
Connor Spiech [1,2,3] ✉, Guilherme Schmidt Câmara[4,5], Julian Fuhrer [6] & Virginia Penhune [1,2,3]

The pleasurable urge to move to music, termed "groove," is thought to arise from the tension between top-down metric expectations or predictions and rhythmic complexity. Specifically, groove ratings are highest for moderately complex rhythms, balancing expectation and surprise. To test this, meter and rhythmic complexity need to be manipulated independently to assess their impact on groove. Thus, we compared Western listeners' ratings for musical clips of varying rhythmic complexity composed in either the most common Western meter (4/4) or less common meters (e.g., 7/8). In several behavioral studies (Experiment 1, N = 143; Experiment 2, N = 120; Experiment 3, N = 120), we used Bayesian regression to show that groove is greatest for moderately complex rhythms, but only in 4/4. In non-4/4 meters, simpler rhythms elicited the greatest groove. This provides support for the theory that bottom-up rhythmic features interact with meter in a way that shapes the pleasurable urge to move to music.

Music can often fill us with a pleasurable urge to move in time with the rhythm. This has been termed "groove" by music psychologists[1,2]. Groove is typically characterized by an inverted U-shaped relationship with rhythmic complexity, such that moderate levels of complexity elicit higher groove ratings than low and high levels of complexity (but see refs. 3–8 for alternative findings). Current theories based on predictive coding propose that this relationship is the result of an interaction between internal models of the musical beat based on experience and rhythmic features of the individual piece of music. Thus, groove arises from both bottom-up rhythmic features and from top-down expectations. Bottom-up features linked to groove include beat salience[2,9,10], pulse clarity[11–14], microtiming[15,16], and syncopation[17–19] which influence the temporal predictability of the music and our ability to synchronize our movements to it[20–26]. These bottom-up properties of the rhythm, however, also interact with listeners' top-down musical expectations based on long term exposure. These top-down expectations are theorized to arise through auditory statistical learning, the innate ability to extract probabilistic features from the immediate environment that are crucial for the elaboration of perceptual models[27–29]. These expectations in the form of song familiarity or musical style preference have been shown to predict a stronger groove response[2,6,30]. Together, this implies that top-down models, not just rhythmic features, contribute to groove because they determine how we internally organize and categorize incoming auditory information when listening. Despite the fact that musical exposure

and enculturation are known to affect groove, the majority of studies have only assessed responses to rhythms based on the most common metrical structures in Western music. Therefore the present study directly tested the interaction between bottom-up rhythmic features and top-down metric structure by comparing groove ratings for musical clips of varying rhythmic complexity in either a common Western meter (i.e., 4/4) for which our participants were very likely to have a strong top-down model, or to more uncommon meters (e.g., 7/8, 5/4, and 9/8) for which we presumed they would have a weaker model according to the literature[31–34].

According to current models based on predictive coding, the tension between our existing metric model and the complexity an individual rhythm we hear explains the inverted U-shaped relationship typically observed between rhythmic complexity and subjective ratings of groove[12,17,19,35–39]. Specifically, at low levels of rhythmic complexity, the metric model goes unchallenged as many musical notes occur on the beats predicted by the meter and thus there is no tension to resolve. At moderate levels of rhythmic complexity, however, some notes deviate from their expected metric positions, challenging the predictions of the model. Here, the metric model can be both maintained and tested by synchronizing real or simulated movements to the beat. The greater pleasure experienced at moderate levels of complexity is then thought to result from the tension that arises from metric predictions and surprising musical notes that violate them. At very high levels of rhythmic complexity, a metric model either cannot be generated or

[1]Department of Psychology, Concordia University, Montreal, QC, Canada. [2]Montreal Centre for Brain, Music and Sound (BRAMS), Montreal, QC, Canada. [3]Centre for Research in Brain, Language and Music (CRBLM), Montreal, QC, Canada. [4]RITMO Centre for Interdisciplinary Studies in Rhythm, Time and Motion, University of Oslo, Oslo, Norway. [5]Department of Musicology, University of Oslo, Oslo, Norway. [6]Centre for Precision Psychiatry, Division of Mental Health and Addiction, Institute of Clinical Medicine, University of Oslo, Oslo, Norway. ✉e-mail: connorrichard.spiech@concordia.ca

fails entirely, resulting in neither tension nor surprise. Crucially, these predictive models of meter are understood to be *encultured*, with listeners learning the metrical frameworks common in their environment through exposure and statistical learning processes[31,32,33,40–42]. Following this logic, we hypothesized that listeners would not experience the tension and surprise that lead to pleasure and the urge to move for moderately complex rhythms in meters outside of 4/4 for which they likely have a weaker top-down model.

Since the 1960's, Western popular music has been dominated by the use of simple quadruple meter (i.e., 4/4 or "common time") where rhythms are grouped into four main beats with binary subdivisions, compared to quadruple meters with ternary subdivisions (e.g., 12/8) as well as triple and duple meters with three or two beats grouped into binary subdivisions (e.g., 3/4 and 6/8)[43]. Complex, additive, and non-isochronous meters (e.g., those with a prime number of pulses or asymmetrical subdivisions) are even less common in Western music[44] and Westerners perform worse at both perception and production tasks using these meters relative to 4/4[25,34,45–47]. Yet so far, most studies on groove have only used stimuli in 4/4, perhaps explaining the term's origin in funk music where common time is ubiquitous[48].

Thus, to investigate the impacts of metric structure and rhythmic complexity on the pleasurable urge to move to music, we asked Western listeners to rate their perceived groove for 63 clips of real, commercially recorded music selected and validated in consultation with two expert musicologists. Half of the clips were in 4/4 and half in one of the less common non-4/4 meters. Both 4/4 and non-4/4 clips spanned a range of rhythmic complexity as measured by pulse entropy calculated from the Music Information Retrieval Toolbox[49,50]. Pulse entropy estimates the predictability of the beat in a musical piece by building an autocorrelation curve of the detected note onsets over time and then calculating the Shannon entropy of the curve's peaks. Importantly, unlike other metrics it can be calculated directly from the waveform of real, polyphonic music. In Experiment 1, we validated the pulse entropy metric by asking participants to rate the perceived rhythmic complexity of each of the musical clips. Pulse entropy was highly correlated with listener ratings of rhythmic complexity for clips in both 4/4 and non-4/4 meters. This confirmed that we could successfully manipulate rhythmic complexity orthogonally to meter with our stimulus set. In Experiment 2, we tested our central hypothesis that listeners would no longer experience the greatest urge to move and pleasure for moderate complexity rhythms if those rhythms were in a meter for which they likely had a weaker top-down model. Finally, in Experiment 3, we tested whether the results of Experiment 2 could be driven by local comparisons between the 4/4 and non-4/4 meters when the excerpts were rated together. That is to say that we wanted to investigate whether metric expectations from the previous trial could influence the perception and subsequent rating of the music in the current trial, i.e., metric priming[51–55]. To accomplish this we had a new group of participants rate the clips in separate, covert blocks of 4/4 and non-4/4 clips and compared them to the ratings of our randomly presented clips in Experiment 2.

## Methods
None of the experiments described in this manuscript were preregistered.

### Participants
A total of 383 participants (mean age = 28.2, SD = 8.8, range: 18–65, 0.3% missing; self-reported gender: 45.2% women, 50.4% men, 4.44% non-binary) were recruited across all three experiments via Prolific and word of mouth. All participants provided informed consent in compliance with the ethical approval granted by the Department of Psychology's University Human Research Ethics Committee at Concordia University (certification number 30018243). Participants were compensated with 7 GBP per hour (~$9) for their participation. A total of 143 people participated in Experiment 1 (mean age = 29.0, SD = 10.0, range: 18–65, 0.7% missing; self-reported gender: 45.5% women, 51.7% men, 2.8% non-binary or gender non-conforming), while 120 people participated in Experiment 2 (mean age = 27.0,

SD = 6.9, range: 18–53; self-reported gender: 48.3% women, 50.0% men, 1.7% non-binary or gender-fluid). For Experiment 3, another 120 participants (mean age = 28.3, SD = 8.8, range: 18–62; self-reported gender: 47.3% women, 50.8% men, 1.7% non-binary or no gender) were recruited and aggregated into the dataset for Experiment 2 to determine whether there was an effect of stimuli presentation (Random or Blocked). To ensure that participants had similar exposure to 4/4 and non-4/4 meters, we only recruited participants from Western Europe, the United States, Canada, Australia, and New Zealand[31,32].

Of our 383 participants, 244 reported receiving some formal musical training with 118 reporting that they still play an instrument. Of the participants who reported formal musical training, the average age that it started was 10.14 (range: 3–40, SD: 4.72 years), the average age that music playing stopped was 19.48 (range: 8–59, SD: 8.97 years), and 108 participants reported playing their instrument for 30 min or more per week (mean: 4.56, range: 0.50–35, SD: 5.50 h). A composite score of total years of musical training was not calculated because some participants reported the age that they stopped playing music without reporting the year that they started playing. This was particularly common among evidently self-taught musicians who still reported playing music but did not report any formal musical training, which made it impossible to estimate how many years they had been actively playing music from the data we collected. Thus, we thought it was safer to simply report the data that we could confidently assess, like the age they started formal training (if they had any), the age they reported stopping their music playing (if they did indeed stop playing), the amount of hours per week spent playing music, and the composite Barcelona Musical Reward Questionnaire score.

Participants' whose rhythmic complexity or groove ratings did not span at least two and a half standard deviations would be excluded from the analysis for non-compliance since variability this low would imply that they were barely using the rating scale and likely not paying attention to the stimuli. However, this was not the case for any participants and thus none were excluded for this reason.

### Musical clips
Musical clips were selected on the following criteria: tempo within (or manipulated to be within if such manipulation didn't induce noticeable auditory distortions/artifacts) 110–130 bpm because past work has shown it's the preferred range for musical tempi and human movement[56,57]. Additionally, the clips needed to be at least four bars long so that a meter could be recognized (lasting ~7–12 s) and needed to contain at least three instrumental layers/streams (one percussive, one bass, and one melodic) and no vocals. Vocals were excluded because they capture attention more than other instruments[58,59], are often highly syncopated[60], could induce unwanted gender effects[61–63], are subject to great individual differences in musical taste[64–66], and because lyrical content impacts emotions and the urge to move[67,68]. The Pulse Entropy of each clip was calculated using the MIR Toolbox's *mirpulseclarity* function with the 'EntropyAutocor' setting as in previous work[10–14,26,37,49,69,70]. In short, this function reads in the audio file, detects the event onsets, runs a sliding autocorrelation window on the detected onsets to construct a curve to determine how self-similar the timing between events is over time, and then calculates the entropy of the peaks of this curve. The average pulse entropy of clips in 4/4 ($M = 0.685$, $SE = 0.016$) and non-4/4 ($M = 0.702$, $SE = 0.012$) meters did not differ ($BF = 0.366$).

The meter of each clip was assessed and labeled by two professional musicologists (author GSC and Dr. Anne Danielsen); 34 were in 4/4 meter and 29 were in meters outside of 4/4. As mentioned in the Introduction, meter is generally considered to be a subjective phenomenon, and therefore any given physical rhythm can potentially be heard as being organized by a variety of meters. Therefore, assigning a notation-based meter to any piece of music is an interpretative endeavor. Nonetheless, based on conventions from Western popular performance and compositional practices, we were able to confidently classify the tracks into 4/4 and non-4/4, despite some of the non-4/4 clips possessing more ambiguous time signatures.

The final selection of 63 musical clips spanned several genres including rock, indie, funk, metal, and electronic music with exemplars from each genre across the entire range of Pulse Entropy values in both Common and Uncommon Meters. As noted in the Results section, several musical clips were excluded from the analyses because they were too familiar or deviated substantially in terms of perceived complexity relative to what was predicted given their pulse entropy values. While the exclusion of these clips did not significantly affect our results, it does beg the question of why participants rated these clips with moderate pulse entropy as being very rhythmically simple. One feature that all three of these clips have in common is that the drum patterns are all quite simple and unsyncopated. Past research has shown that timing information is encoded better from low frequencies and percussive instruments[71–73]. Thus, participants might have based their ratings on the simpler rhythms in the lower frequency, percussive instruments rather than the higher frequency instruments with more rhythmic ornamentation that the MIR Toolbox captured. Given that these lower frequencies have been linked to groove[10,74,75], future work is being planned to investigate this possibility directly by manipulating the rhythmic complexity of different instruments in relation to each other. More information about the musical clips is detailed in Table 1 of the Supplementary Materials.

## Procedure

Each experimental session began with an informed consent screen followed by a brief survey about their musical background. Afterwards, participants read instructions clarifying the nature of the experiment and the meanings of the different rating scales. Ratings were collected via dichotomous visual analog scales. Specifically, for Experiment 1 we defined Perceived Complexity as "how rhythmically complex the music was" (with poles labeled "The rhythm was very simple" vs. "The rhythm was very complex"), Predictability as "how predictable the timing of each musical note was" (poles: "I had no idea when the next sound would occur" vs. "I knew exactly when the next sound would occur"), and Pulse Clarity as "how easy would it be to move in sync with the music" (poles: "It would be very difficult" vs. "It would be very easy"). We further instructed participants to base their ratings of complexity and predictability on the music's rhythm and not other aspects like its melody, harmony, or timbre. In Experiments 2 and 3, Urge to Move was defined as "how much the music made you want to move" (poles: "I did not want to move at all" vs. "I wanted to move a lot") with the clarification that wanting to move can be "as simple as feeling the urge to tap your foot or nod your head to the music and does not necessarily mean that you wanted to dance to it" and Pleasure as "how much you enjoyed the musical clip" (poles: "I did not enjoy it at all" vs. "I enjoyed it a lot"). In all three experiments, Familiarity was defined as "how familiar you were with that song" (poles: "I am certain that I have never heard that song before" vs. "I am certain that I know that song") and Style Preference as "how often you listen to that style of music" (poles: "I never listen to this style of music" vs. "I often listen to this style of music") to be used as random effects since previous work has shown they substantially impact Urge to Move and Pleasure[6,30]. Participants were instructed to use the best headphones they had available and to try and use the entire scale because they would not be paid if they simply clicked through the experiment without truly rating the clips. Participants were then presented with each musical clip and subsequently rated it using the different visual analog scales before pressing another button to advance to the next trial. In Experiments 1 and 2, musical clips were presented randomly with Common and Uncommon meters interspersed among each other whereas in Experiment 3, the clips were blocked with half of the participants listening to and rating all of the Common Meter clips first while the other half listened to and rated all of the Uncommon Meter clips first. Finally, subjects completed the Barcelona Musical Reward Questionnaire[76] before being directed back to Prolific for their payment. All experiments were programmed in jsPsych[77] and hosted on Pavlovia.

## Behavioral data processing and analysis

To address our research questions, we employed hierarchical Bayesian regression using the 'rstanarm' package[78] in R (version 4.1.2)[79]. All models were estimated using Markov chain Monte Carlo sampling (No-U-Turn Sampler variant of Hamiltonian Monte Carlo algorithm) with six chains of 5000 iterations and a warmup of 2000 samples using the default weakly informative priors. QR decomposition was employed for models with multiple predictors.

In all three experiments, ratings of Familiarity and Style Preference were modeled as by-subject random effects while the remaining ratings were used as the dependent variables. Pulse Entropy and Meter were treated as fixed effects with random slopes for Pulse Entropy to account for individual differences. Pulse Entropy was modeled linearly in Experiment 1 to validate that our musical clips reliably tracked perceived rhythmic complexity as intended. In Experiments 2 and 3, orthogonal linear and quadratic slopes were estimated for Pulse Entropy based on the groove literature[17,19,37,39,80].

Potential covariates were selected using a Pearson's correlation matrix of all measured variables in an experiment. Only questionnaire items that were significantly correlated with dependent variables after Bonferroni correction[81] were included in the models. For Experiment 3, an additional fixed effect of Context corresponding to presentation order (Random or Blocked) was added. The best-performing model was then selected using Pareto-smoothed importance sampling leave-one-out (PSIS-LOO) cross-validation[82] with a k-threshold of 0.7. The resulting expected log-predictive densities (ELPDs) were then used to select the model with the best predictive performance. This is a form of model selection like using the Akaike information criterion (AIC) to estimate the best-fitting model without overfitting the data commonly employed in frequentist statistics. However, PSIS-LOO cross-validation provides the additional advantage of standard errors for each model, allowing us to better quantify how much better one model is than another. Finally, Bayes factors for each parameter were extracted from the best model by computing the Savage-Dickey density ratio using the 'bayestestR' package[83] to assess the significance of each predictor. As described in the results section, these allow us to quantify the degree of evidence for or against each parameter differing from its null hypothesis (in this case, our weakly informative priors). In frequentist terms, these can be thought of as roughly analogous to p-values that can additionally quantify the evidence *against* an effect, rather than simply not observing one. We then report point estimates corresponding to the median of the posterior distribution for these parameters along with their 95% credible intervals computed from the highest density interval of the posterior. Point estimates and credible intervals are essentially Bayesian equivalents of beta coefficients and their confidence intervals obtained in frequentist regressions.

## Reporting summary

Further information on research design is available in the Nature Portfolio Reporting Summary linked to this article.

## Results

In the following subsections, we present the results of each experiment. Bayesian analyses to test the directionality of findings – in favor of or against the null hypothesis – are used throughout. These analyses are critical for us to test our hypotheses in Experiment 1 where we need to quantify the evidence against an interaction between perceived rhythmic complexity and meter, and in Experiment 3 where we want to quantify the evidence against the effect of context. For readers unfamiliar with the interpretation of Bayesian analyses, expected log-predictive densities (ELPD) can be analogized to Akaike or Bayesian information criteria used for frequentist model selection except with standard errors to better quantify how much better one model predicts the data over another. Bayes factors can be thought of as p-values that can additionally assess evidence against the null hypothesis. Bayes factors close to one indicate no evidence in favor of either the null or experimental hypothesis; specifically, a Bayes factor of exactly one indicates that the experimental hypothesis is equally likely as the null hypothesis. Bayes factors between three and ten are typically considered moderate evidence for the experimental hypothesis, while their reciprocals (one third and one tenth, respectively) are considered moderate evidence for the null hypothesis. Bayes factors greater than ten (or less than one tenth) are

**Fig. 1 | The results of Experiment 1.** Perceived Rhythmic Complexity increases with pulse entropy while Predictability and Pulse Clarity decrease with pulse entropy, $N = 143$.

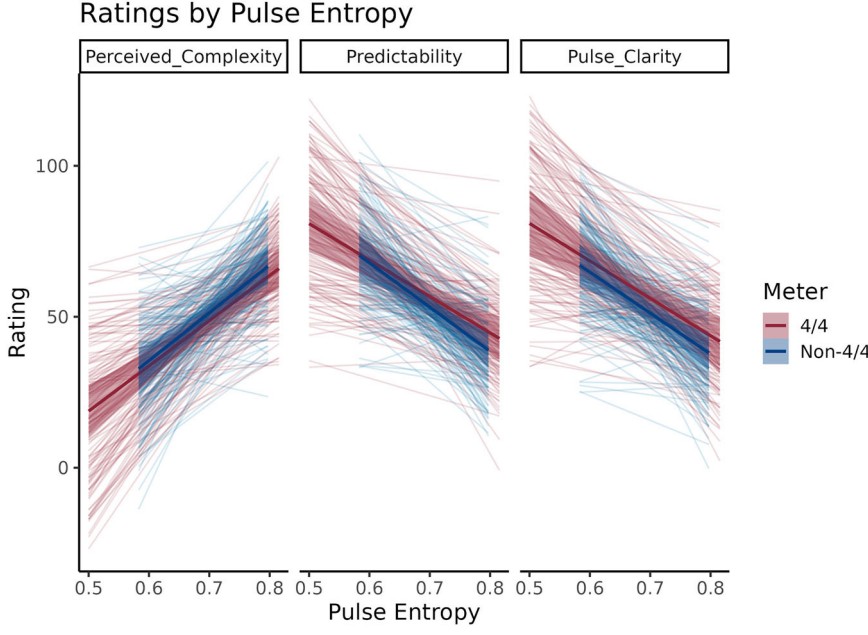

considered strong evidence for the experimental or null hypothesis respectively[84].

## Experiment 1 - Rhythmic complexity ratings increase with pulse entropy regardless of meter

Here, we present the results of Experiment 1 where we sought to confirm that the rhythmic complexity ratings of our musical clips corresponded to measured pulse entropy values regardless of meter. Specifically, participants rated the perceived complexity of the music's rhythm, the predictability of each note's timing, and the pulse clarity (how easy it would be to move to the music). Additionally, song familiarity and style preference were rated as well to be used as random effects to control for their known influence on pleasure and the urge to move to music (see Methods for more details)[6,30]. We first report the inclusion of a significant covariate, followed by model selection and results.

One musical experience question from our questionnaire, Age Music Playing Stopped, was significantly correlated with Predictability ($r(141) = 0.340$, 95% CI [0.180, 0.474], $p = 0.003$) and Pulse Clarity ($r(141) = 0.383$, 95% CI [0.227, 0.512], $p < 0.001$) and was thus added to these models as a covariate interacting with Pulse Entropy and Meter. The full correlation matrix with all recorded responses is depicted in Supplementary Fig. 1. All models converged as evidenced by R-hat values of 1.0 and Pareto's k estimates all fell under 0.7, indicating no outlying observations.

For Perceived Complexity, the model with predictors for Meter and Pulse Entropy had the best predictive performance, with ELPD differences two standard errors greater than the model with Pulse Entropy only and roughly 18 times greater than the Meter only and intercepts only models. The Bayes factors for this model revealed a large positive effect of Pulse Entropy ($\beta_1 = 152.71$, 95% CI [140.04, 165.69], BF $= 5.85 \times 10^{25}$) with no effect of Meter ($\beta_1 = 1.67$, 95% CI [-9.70, 12.17], BF $= 0.004$) or its interaction with Pulse Entropy ($\beta_1 = -6.22$, 95% CI [-21.94, 9.20], BF $= 0.005$). Thus, participants perceived musical clips with greater Pulse Entropy as being more rhythmically complex regardless of their metric structure.

For Predictability, the model with predictors for Meter, Pulse Entropy, and Age Music Playing Stopped displayed the best predictive performance, with ELPD differences about two, three, 14, and 15 standard errors greater than the models with Pulse Entropy and Meter, Pulse Entropy alone, Meter alone, and random intercepts only, respectively. Extracted Bayes factors revealed a large negative relation between Pulse Entropy and Predictability ($\beta_1 = -120.99$, 95% CI [-141.61, -101.30], BF $= 2.98 \times 10^{20}$) with no effect of

Meter ($\beta_1 = -22.89$, 95% CI [-40.01, -6.13], BF $= 0.130$), Age Music Playing Stopped ($\beta_1 = 0.52$, 95% CI [-0.17, 1.17], BF $= 0.012$), or any interactions (Pulse Entropy x Meter: $\beta_1 = 35.32$, 95% CI [10.60, 58.91], BF $= 0.216$; Pulse Entropy x Age Music Playing Stopped: $\beta_1 = -0.43$, 95% CI [-1.35, 0.50], BF $= 0.006$; Meter x Age Music Playing Stopped: $\beta_1 = 0.66$, 95% CI [-0.11, 1.45], BF $= 0.015$; Pulse Entropy x Meter x Age Music Playing Stopped: $\beta_1 = -0.80$, 95% CI [-1.89, 0.34], BF $= 0.009$). Note that because the poles on the Predictability rating scale ranged from "I had no idea when the next sound would occur" to "I knew exactly when the next sound would occur", the direction of this effect is the opposite of and consistent with what we found for Perceived Complexity.

For Pulse Clarity, the model with Meter, Pulse Entropy, and Age Music Playing Stopped exhibited the best predictive performance, with ELPD differences roughly three, five, 14, and 15 standard errors greater than the Pulse Entropy and Meter, Pulse Entropy only, Meter only, and intercepts only models, respectively. Similar to the Predictability model, the Bayes factors retrieved from this model show a strong negative slope for Pulse Entropy ($\beta_1 = -103.58$, 95% CI [-123.24, -83.41], BF $= 2.11 \times 10^9$) regardless of Age Music Playing Stopped ($\beta_1 = 0.64$, 95% CI [-0.05, 1.30], BF $= 0.021$) or Meter ($\beta_1 = -11.64$, 95% CI [-28.38, 5.00], BF $= 0.009$), with no interactions (Pulse Entropy x Meter: $\beta_1 = 18.49$, 95% CI [-4.74, 42.73], BF $= 0.011$; Pulse Entropy x Age Music Playing Stopped: $\beta_1 = -0.59$, 95% CI [-1.52, 0.33], BF $= 0.008$; Meter x Age Music Playing Stopped: $\beta_1 = 0.61$, 95% CI [-0.22, 1.37], BF $= 0.011$; Pulse Entropy x Meter x Age Music Playing Stopped: $\beta_1 = -0.58$, 95% CI [-1.72, 0.54], BF $= 0.006$). This indicates that participants rated increasingly entropic clips as being more difficult to move to regardless of both the clips' meters or when they stopped playing music.

Results from all three models are displayed in Fig. 1 below. While these results validated pulse entropy as a measure of rhythmic complexity regardless of meter with high reliability (Cronbach's $\alpha = 0.827$ [0.819, 0.835]), several clips had to be excluded from this and all subsequent analyses for excessive familiarity ratings (Billie Jean by Michael Jackson, Seven Nation Army by The White Stripes, Beat It by Michael Jackson, and Mission Impossible by Lalo Schiffrin Orchestra were all rated >92/100 at the group average). Furthermore, three additional tracks (1992 by no_4mat, Some Kind of Game by Against All Logic, and Walk by Pantera) were excluded for ratings that excessively deviated from predicted values based on their Pulse Entropy (potential reasons for why they deviated so much are discussed in the Methods). Including all seven of these tracks did not substantially alter the pattern of results in any of our experiments, and in Experiments 2 and 3,

**Fig. 2 | Urge to Move and Pleasure ratings in Experiment 2 by Pulse Entropy and Meter where musical clips were presented in a randomly interspersed order regardless of Meter, $N = 120$.** Like previous studies manipulating Rhythmic Complexity in Common meters, Urge to Move and Pleasure exhibited a significant inverted U-shaped relationship with Pulse Entropy. In Uncommon meters, however, this relationship with Urge to Move flipped while there was very little evidence for any relationship with Pleasure.

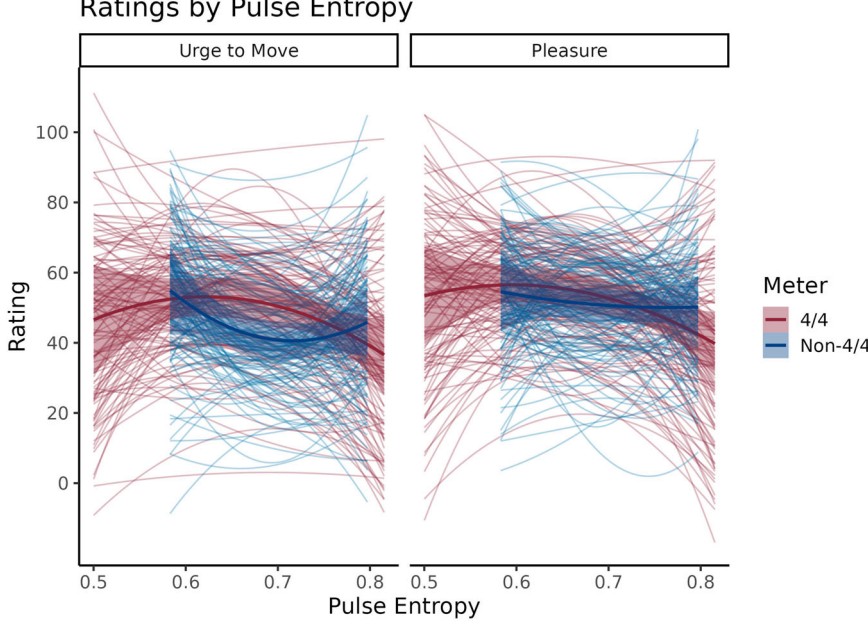

including these items actually exaggerated the inverse quadratic effects for Common Meters.

**Experiment 2 - Groove ratings differ for 4/4 and non-4/4 meters**
Experiment 2 investigated Urge to Move and Pleasure ratings' relation to pulse entropy in common vs. uncommon meters. Specifically, we expected that the inverted U-shaped relationship would only be present in Common Meters. Results are reported first for the models of Urge to Move ratings and then for Pleasure ratings.

No questionnaire responses were significantly correlated to Urge to Move or Pleasure ratings (see Supplementary Fig. 2) and so we only compared models with predictors for Meter, Pulse Entropy (as orthogonal linear and quadratic slopes), their interaction, and random intercepts. All models converged as evidenced by R-hat values of 1.0, however, several models had Pareto's k estimates that exceeded our threshold of 0.7 (from one to four problematic observations). Because there were so few problematic observations, we refit the models with each problematic observation left out one at a time to compute their ELPD contributions directly before combining them with the results from the rest of the cross-validation as suggested by the authors of the 'loo' package[82].

The models with Meter and Pulse Entropy had the best predictive performance for both Urge to Move and Pleasure ratings, with ELPD differences that were about five to seven (for Urge to Move) and three to four standard errors (for Pleasure) greater than those of the Meter only, Pulse Entropy only, and intercepts only models. Bayes factors from the Urge to Move model revealed evidence for Pulse Entropy's linear ($\beta_1 = -184.58$, 95% CI [-269.75, -101.20], BF = 10.95) and quadratic terms ($\beta_1 = 346.62$, 95% CI [252.34, 436.98], BF = $3.34 \times 10^4$), Meter ($\beta_1 = 3.46$, 95% CI [2.45, 4.47], BF = $3.80 \times 10^3$), and an interaction between Meter and the quadratic term ($\beta_1 = -508.49$, 95% CI [-614.07, -407.24], BF = $3.84 \times 10^8$) but not the linear term ($\beta_1 = 68.88$, 95% CI [-22.49, 163.95], BF = 0.009). Follow-up models demonstrated that this interaction was driven by a negative quadratic slope for Pulse Entropy in Common Meter clips ($\beta_1 = -131.67$, 95% CI [-169.62, -90.56], BF = $2.26 \times 10^3$) and a positive quadratic slope for Pulse Entropy in Uncommon Meter clips ($\beta_1 = 152.09$, 95% CI [112.33, 190.31], BF = $1.00 \times 10^6$). For Pleasure, the Bayes factors only showed evidence for Meter's interaction with the quadratic ($\beta_1 = -175.22$, 95% CI [-262.30, -88.84], BF = 3.49) and linear terms of Pulse Entropy ($\beta_1 = -179.56$, 95% CI [-257.48, -100.40], BF = 10.25), respectively, while there was no evidence for Pulse Entropy (Linear: $\beta_1 = 74.20$, 95% CI [6.32, 144.42], BF = 0.028;

Quadratic: $\beta_1 = 58.56$, 95% CI [-19.68, 134.11], BF = 0.009) or Meter ($\beta_1 = 0.22$, 95% CI [-0.68, 1.07], BF = 0.003) on their own. The follow-up models here revealed that this interaction arose from negative linear ($\beta_1 = -79.04$, 95% CI [-117.39, -40.22], BF = 3.49) and quadratic slopes in Common Meter clips ($\beta_1 = -92.39$, 95% CI [-126.67, -58.39], BF = 474.45) that were absent in clips with Uncommon Meters (Linear: $\beta_1 = 52.94$, 95% CI [18.53, 86.21], BF = 0.400; Quadratic: $\beta_1 = 29.79$, 95% CI [-2.69, 63.13], BF = 0.020).

Altogether, these results indicate that moderate complexity rhythms evoked the greatest urge to move and pleasure *only* for 4/4 meters. Non-4/4 meters generated slightly lower ratings of urge to move overall that were greatest for low rhythmic complexity. Pleasure ratings did not credibly differ across levels of complexity for non-4/4 meters. This analysis is depicted in Fig. 2.

**Experiment 3 - Groove ratings for non-4/4 meters are not dependent on presentation context**
In Experiment 2, 4/4 and non-4/4 clips were presented together in randomized order. Therefore, Experiment 3 was conducted to ensure that participants' ratings from Experiment 2 were not influenced by the meter of a previously presented clip. To accomplish this, we recruited an additional sample in which listeners rated clips in covert blocks of 4/4 and non-4/4 that were counterbalanced across participants. The results of this sample were then compared with those from Experiment 2. An effect of Context (Blocked vs. Random) would indicate that participants were *not* rating each clip independently of each other. Again, we first report the results of the Urge to Move models and then the results of the Pleasure models.

Similar to Experiment 2, no questionnaire items were significantly correlated with any of our dependent variables (see Supplementary Fig. 3) and so only Context was added to the models to confirm that presenting stimuli in blocks of all Common or all Uncommon Meters was no different than randomly interspersing them. All models converged as evidenced by R-hat values of 1.0, however, several models had Pareto's k estimates that exceeded our threshold of 0.7 (from one to four problematic observations). We dealt with these estimates in the same fashion as for Experiment 2.

For both Urge to Move and Pleasure, the models with only Pulse Entropy and Meter displayed the best predictive performance, outperforming the model that included Context by 2.5 standard errors of the ELPD difference. This indicates that Context did not improve the model's predictions in any significant way; extracting the Bayes factors from this

**Fig. 3 | Urge to Move and Pleasure ratings in Experiment 3 by Pulse Entropy and Meter where musical clips were presented in blocks of only Common or only Uncommon Meters, $N = 240$.** The same pattern of results as Experiment 2 was observed: Urge to Move and Pleasure varied along an inverted U-shaped curve with Pulse Entropy in Common Meters while Urge to Move varied along a leftward-skewed positive quadratic curve with Pulse Entropy and Pleasure in a negative linear fashion for clips in Uncommon Meters.

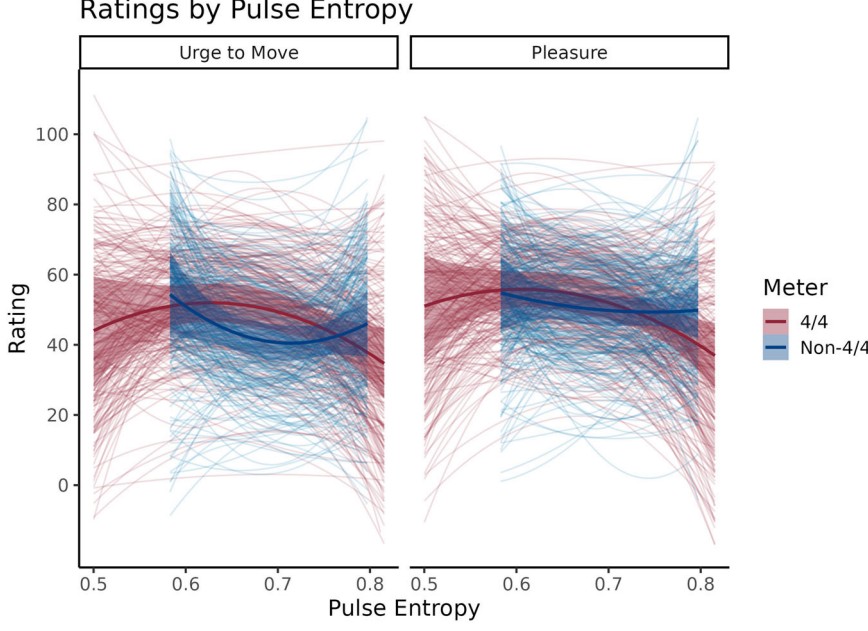

model confirmed that there was extreme evidence against a Context effect (Urge to Move, Context: $\beta_1 = 0.74$, 95% CI [-2.80, 4.14], BF = 0.009; Urge to Move, Context x Linear Pulse Entropy: $\beta_1 = -59.47$, 95% CI [-226.20, 102.07], BF = 0.005; Urge to Move, Context x Quadratic Pulse Entropy: $\beta = 20.91$, 95% CI [-161.26, 207.31], BF = 0.005; Urge to Move, Context x Meter: $\beta_1 = 0.64$, 95% CI [-0.78, 2.17], BF = 0.004; Urge to Move, Context x Linear Pulse Entropy x Meter: $\beta_1 = 34.23$, 95% CI [-151.03, 224.27], BF = 0.003; Urge to Move, Context x Quadratic Pulse Entropy x Meter: $\beta_1 = -10.44$, 95% CI [-216.65, 197.08], BF = 0.002; Pleasure, Context: $\beta_1 = 1.24$, 95% CI [-1.97, 4.28], BF = 0.010; Pleasure, Context x Linear Pulse Entropy: $\beta_1 = 56.87$, 95% CI [-76.62, 190.35], BF = 0.005; Pleasure, Context x Quadratic Pulse Entropy: $\beta_1 = 9.00$, 95% CI [-149.64, 159.47], BF = 0.004; Pleasure, Context x Meter: $\beta_1 = -0.72$, 95% CI [-1.97, 0.56], BF = 0.004; Pleasure, Context x Linear Pulse Entropy x Meter: $\beta_1 = -3.60$, 95% CI [-153.75, 152.84], BF = 0.002; Pleasure, Context x Quadratic Pulse Entropy x Meter: $\beta_1 = 1.30$, 95% CI [-168.76, 182.07], BF = 0.002). The ELPD differences were even more pronounced for the Meter only, Pulse Entropy only, Context only, and intercepts only models which ranged from about seven to nine standard errors greater than their ELPD differences for Urge to Move and about five standard errors greater for Pleasure. The same pattern of results as Experiment 2 emerged from the Bayes factors here: extreme evidence was found for both Pulse Entropy terms (Linear: $\beta_1 = -231.20$, 95% CI [-313.46, -148.96], BF = 154.31; Quadratic: $\beta_1 = 481.74$, 95% CI [387.72, 571.33], BF = $8.85 \times 10^8$), Meter ($\beta_1 = 3.13$, 95% CI [2.41, 3.88], BF = $7.09 \times 10^5$), and their interaction with the quadratic ($\beta_1 = -715.53$, 95% CI [-817.67, -610.25], BF = $8.72 \times 10^9$) but not the linear term ($\beta_1 = 80.77$, 95% CI [-7.81, 179.48], BF = 0.011) for Urge to Move while extreme evidence was found only for the interaction between Meter and Pulse Entropy's linear and quadratic terms (Linear: $\beta_1 = -253.17$, 95% CI [-330.33, -177.35], BF = 897.90; Quadratic: $\beta_1 = -248.57$, 95% CI [-330.95, -159.25], BF = 378.81) but not Pulse Entropy (Linear: $\beta_1 = 77.13$, 95% CI [9.71, 143.73], BF = 0.029; Quadratic: $\beta_1 = 78.02$, 95% CI [5.16, 157.56], BF = 0.016) or Meter ($\beta_1 = 0.57$, 95% CI [-0.03, 1.20], BF = 0.011) alone for Pleasure. The interactions further replicated our results from Experiment 2. Negative linear and quadratic slopes for Pulse Entropy emerged in Common Meter clips for Urge to Move (Linear: $\beta_1 = -102.46$, 95% CI [-141.68, -58.82], BF = 53.05; Quadratic: $\beta_1 = -190.21$, 95% CI [-229.83, -149.28], BF = $1.42 \times 10^7$) and Pleasure (Linear: $\beta_1 = -128.52$, 95% CI [-166.57, -91.35], BF = $1.70 \times 10^3$; Quadratic: $\beta_1 = -134.51$, 95% CI [-169.52, -100.85], BF = $2.85 \times 10^4$) whereas the negative linear slope disappeared ($\beta_1 = -28.75$, 95% CI [-67.10,

13.68], BF = 0.009) and the quadratic slope for Urge to Move reversed polarity ($\beta_1 = 207.59$, 95% CI [168.90, 247.85], BF = $2.90 \times 10^9$) while all effects of Pulse Entropy were absent (Linear: $\beta_1 = 55.94$, 95% CI [22.15, 89.10], BF = 0.545; Quadratic: $\beta_1 = 37.87$, 95% CI [4.62, 71.86], BF = 0.036) for Pleasure. Because it could be argued that our effects are driven by the slightly larger range of pulse entropy values in 4/4, we reduced the number of 4/4 clips to match the range of the non-4/4 clips and re-ran the analyses. Supplementary Analysis 1 documents the details of this control analysis, but in short, this analysis only seems to trim off the decrease in groove for lower complexity stimuli in 4/4 while maintaining elevated groove for mid-complexity stimuli.

In sum, participants did *not* rate our musical clips differently depending on whether they had previously listened to clips of the same or different meters. Instead, they rated them as they had in Experiment 2: along an inverted U-shaped curve in 4/4, and along a leftward-skewed positive quadratic curve for Urge to Move with no clear relationship to pleasure in uncommon meters. This is displayed below in Fig. 3.

## Discussion

In this suite of experiments, we have shown that rhythmic complexity can be manipulated orthogonally to metric complexity and that the pleasurable urge to move to music is greatest for moderately complex rhythms, but only in 4/4, the meter for which our listeners likely had the strongest top-down model. Instead, for non-4/4 meters, the urge to move was greatest for *lower* rhythmic complexity, varying along a skewed positive quadratic while pleasure did not vary with rhythmic complexity. This interaction between rhythmic and metric complexity did not depend on the context in which the musical clips were rated (blocked or randomly interspersed), indicating that this effect is likely related to long-term experience, and is not an artifact of metric priming by previously heard clips. These results align with the predictive coding framework where groove is understood as the result of top-down predictions (derived from long-term experience) interacting with bottom-up information from specific rhythmic stimuli.

Top-down knowledge influences our perception and cognition throughout the nervous system, from color perception[85,86] and visual working memory[87,88] to speech intelligibility[89] and musical syntax[90,91]. Rhythm processing appears to be no exception. Past work has shown that listeners' metric expectations shape their neural responses such that activity is better locked to more strongly predicted beat points than weakly predicted points, even in completely isochronous rhythms[92–94]. Behaviourally,

Western adults also struggle to detect metric violations in meters uncommon in their culture, but common in other musical traditions in Turkey or the Balkans[34,45,95]. Furthermore, rhythmic synchronization and production abilities related to groove also seem to be influenced by enculturation in both laboratory and simulation studies; participants exhibit consistent biases toward and better performance for rhythms organized in ways common in their cultures[25,31,33,40,41,42,96].

Here, we show that metric structure shapes our embodied responses to rhythms of different complexity as indexed by participants' urge to move ratings. Through the lens of the predictive coding account, the inverted U-shaped relationship between the urge to move and rhythmic complexity is the result of tension created by top-down expectations of where the beat will occur and deviations from those expectations in the music. Moderately complex rhythms contain deviations from our expectations and thus real or imagined movement may allow us to clarify or reinforce those expectations[97]. Low complexity rhythms do not violate our expectations and thus do not generate the urge to move, while high complexity rhythms may not allow us to generate strong predictions. Based on these ideas, our finding that the urge to move to music only displays an inverted U-shaped relationship with rhythmic complexity in common meters may be because the metric model for uncommon meters is too weak and thus does not create tension between the rhythm and meter. Consistent with this explanation, urge to move ratings in non-4/4 meters instead reverted to a positive quadratic slope with higher ratings skewed towards lower levels of rhythmic complexity, perhaps because the structure of these rhythms is simple enough to move to without the need for a reinforcing top-down metric model.

While our results confirm past qualitative findings that groove is deeply tied to rhythmic features[2,68,98], the typical coupling to pleasure was only observed within the context of the common 4/4 meter. While we did not see the typical inverted U-shaped function for pleasure with non-4/4 meters, participants did not rate them as less pleasurable overall. This dissociation between pleasure and the urge to move may have arisen because pleasure is more multifaceted, being affected by other properties of music like melodic complexity[99,100], harmonic complexity[101], and timbre[59,102]. Thus, our participants seem to have derived pleasure from other features of the music. Put simply, this suggests that pleasure is more strongly linked to rhythmic features when they fit an existing model; when they don't, other features may be evaluated instead. As suggested by Brattico and colleagues, our understanding of musical pleasure may be better served by more holistic measures of music's many acoustic properties[103].

Finally, it remains an open question whether our results reflect pure statistical learning in Westerner listeners[3] or rather some innate bias towards simpler rhythmic groupings. Future work supporting the effects of enculturation could come from testing people with greater exposure to non-4/4 meters, including cross-cultural samples[32,33] or populations of specially trained musicians[31]. In our study, however, no musicianship-related covariates emerged as significant predictors of perceived rhythmic complexity or groove, implying that the musicians in our samples rated the clips similarly to nonmusicians (see also Supplementary Analysis 3). This could be a result of Western musicians favoring 4/4, but to rule this out definitively, musicians with specific experience playing in non-4/4 meters would have to be recruited.

Alternatively, there is evidence for innate preferences for meters with simple binary subdivisions like 4/4. Researchers have observed that rhythmic, and especially musical, behaviors appear to exhibit a binary bias, both in the standard Western repertoire[104,105] and in diverse musical traditions studied worldwide[106]. For instance, rhythm perception and production studies demonstrate clear biases for rhythms with these simple integer ratios in both Western adults[107–109] and children as young as five or six years of age[110–112], as well as in most cultures that have been studied so far, even those whose music incorporates more complex meters like 7/8 and 5/4[31,32,33]. Regarding groove specifically, recent developmental work has shown that three to seven year old children move more to moderately complex rhythms and high groove music, implying that they are already sensitive to these temporal features, even if their synchronization abilities aren't as precise as those of adults[39,113]. If a bias for binary meters is truly universal, it could arise from some biological constraint on a limited cognitive resource[114–116], through a shared motor behavior like bipedalism that influences metric preferences[117,118], or a physical affinity for neural populations to attune and resonate to simpler integer ratios[119–121].

## Limitations
There are several limitations to this work. First, as discussed above, cross-cultural research could disentangle potential effects of statistical learning from the binary bias and more definitively rule in favor of one interpretation over the other. Our results, as a first step into investigating the influence of meter and rhythmic complexity on groove, cannot accomplish this alone. Second, we did not have a direct measure of individuals' metric expectations for each track. Creating such a measure would be very useful for future studies, however, it would require extensive development to design a test that works for non-musicians who do not have a formal conception of meter.

## Conclusion
In conclusion, these results provide evidence that bottom-up rhythmic features interact with top-down metric structures in a way that shapes the pleasurable urge to move to music. Consistent with the predictive coding framework, when metric expectations are stronger, moderately complex rhythms generate tension that elicits a pleasurable urge to move along in a way that tests and reinforces those expectations. When these metric expectations are weaker or absent, no tension is generated and the urge to move only arises for simpler rhythms and pleasure is decoupled from rhythmic features. In this way, the groove experience is not merely the sum of rhythmic events, but rather a musical gestalt comprised of both the rhythm and our metric interpretation of that rhythm.

## Data availability
Data for all analyses can be accessed at the following Open Science Framework repository: https://osf.io/6an7j/?view_only=da98815674c8452fad18a56699eb7ed8.

## Code availability
Code for all analyses can be accessed at the following Open Science Framework repository: https://osf.io/6an7j/?view_only=da98815674c8452fad18a56699eb7ed8.

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

## Acknowledgements

We would like to thank Alexander Szorkovsky for his helpful recommendations of several musical clips, Calder Hannan for his advice on musicological terminology, Alberto Ara for his statistical consultations, and Justin London for his constructive discussion on the binary bias. We would also like to extend a huge thanks to Anne Danielsen for verifying the meters of the different clips as suggested during review. Finally, we would like to thank Laura Cirelli for her helpful comments and suggestions during the review process. This work was partially supported by the Research Council of Norway through its Centres of Excellence scheme, project number 262762, by Virginia Penhune's National Sciences and Engineering Research Council of Canada grant (NSERC 2021-04026), and the Concordia University Horizon Postdoctoral Fellowship. The funders had no role in study design, data collection and analysis, decision to publish or preparation of the manuscript.

## Author contributions

All authors were involved with the conceptualization of the study and its experimental design, with the initial idea coming from author C.S. Stimuli were selected and prepared by authors C.S. and G.S.C. while author JF wrote the experiment script for online data collection. Data curation, analysis, and visualization were done by author CS with feedback from author J.F. C.S. wrote the first draft of the manuscript, all authors revised the manuscript, and author VP supervised the entire study.

## Competing interests

The authors declare no competing interests.
