## [Transparent Peer Review file · Communications Psychology]

4/4 and More: Rhythmic complexity more strongly predicts groove in common meters

Corresponding Author: Dr Connor Spiech

Version 0:

Decision Letter:

Dear Dr Spiech,

Thank you for your patience during the peer-review process. Your manuscript titled "4/4 and More: Groove in Uncommon Meters" has now been seen by 3 reviewers, whose comments are appended below. You will see that they find your work of some potential interest. However, they have raised quite substantial concerns that must be addressed. In light of these comments, we cannot accept the manuscript for publication, but would be interested in considering a revised version that fully addresses these serious concerns.

We hope you will find the Reviewers' comments useful as you decide how to proceed. Should additional work allow you to address these criticisms, we would be happy to look at a substantially revised manuscript. If you choose to take up this option, please highlight all changes in the manuscript text file, and provide a detailed point-by-point reply to the reviewers.

Editorially, we consider it important that the revised manuscript stays close to the data in its interpretation of the findings. You will see that both Reviewers 1 and 3 encourage you to run the experiment with participants who have greater exposure to non-4/4 stimuli. We also encourage you to conduct a replication in a Turkish or Bulgarian population (or any other suitable population). This type of data would allow you to make broader claims regarding the top-down nature of groove should you choose to undertake it. In the absence of this data, please remove your interpretations regarding top-down processing and the influence of culture as well as acknowledge alternative interpretations (e.g., groove functions differently in non-4/4 music).

Furthermore, please address the reviewers' concerns regarding other features of the music, especially tempo, familiarity, and pulse entropy, as well as the concerns regarding the analysis of musical background. If Experiment 2 and Experiment 3 were separate data collections than please continue to report them separately.

I am attaching a checklist that details critical reporting requirements for the revised manuscript. Please attend to each item and ensure your manuscript is fully compliant. We are requesting that your manuscript aligns with these requirements as this facilitates the evaluation of your manuscript, reducing delays in re-review and potential future acceptance. If your revised manuscript is not aligned with these requests on major issues, such as those concerning statistics, it may be returned to you for further revisions without re-review. Additional information can be found in our style and formatting guide <https://www.nature.com/documents/commspsychol-style-formatting-guide-accept.pdf> Communications Psychology formatting guide.

If the revision process takes significantly longer than five months, we will be happy to reconsider your paper at a later date, provided it still presents a significant contribution to the literature at that stage.

Please use the following link to submit your

- revised manuscript,
- point-by-point response to the referees' comments,
- cover letter (as a separate document),
- the Editorial Policy Checklist (see below),
- the Reporting Summary (see below), and
- the completed Editorial Request Table (attached):

Link Redacted

Thank you for the opportunity to review your work.

Best regards,

Jennifer Bellingtier

Jennifer Bellingtier, PhD
Senior Editor
Communications Psychology

REVIEWER EXPERTISE:

Reviewer #1 groove, rhythmic complexity, pulse entropy
Reviewer #2 groove, rhythmic complexity, pulse entropy
Reviewer #3 groove, rhythmic complexity, pulse entropy

REVIEWER REPORTS:

Reviewer #1 (Remarks to the Author):

The present manuscript reports three experiments examining the relationship between rhythmic complexity, groove, and meter. English-speaking participants were tested on-line and presented with clips of music in 4/4 meter or in non-4/4 meter and rated these clips for rhythmic complexity (Experiment 1), and groove and pleasure (Experiment 2-3). The results replicate existing findings that the relationship between rhythmic complexity and groove in 4/4 music is characterized by an inverse U function with moderate complexity rhythms rated highest in groove and pleasure. By contrast, groove ratings of the non-4/4 musical clips were generally characterized by a positive quadratic function suggesting relatively lower and higher complexity rhythms were rated groovier by listeners. Given the assumption that the listeners were unfamiliar with non-4/4 meters, the authors argue that they provide evidence that a listeners' familiarity with the metrical structure of music influences their experience of groove.

The questions behind this study are fascinating and I love the idea of studying how listeners experience groove in music with non-isochronous meters because. As the researchers point out, this has not yet been studied and it addresses some important theoretical questions about why we experience groove, and the relative importance of listener's cultural experience vs. the structure of the music. Even though I wanted to love this paper, I am not at all convinced that this evidence provides support for the notion that "top-down models of metrical structure shape our embodied responses to rhythms of different complexity as indexed by participants' urge to move ratings" as the authors claim.

Although the authors somewhat acknowledge this limitation in the discussion, I think it is a serious problem that there is no comparison group with greater exposure to non-isochronous meters. In my view it seems equally if not more likely that the observed differences between ratings of isochronous vs non-isochronous stimuli arise from features of this particular non-isochronous stimulus set and/or perhaps because (maybe?) groove functions differently in non-isochronous than in isochronous music.

Ideally, the design of this study would compare groups of individuals with and without exposure primarily to non-isochronous meters (perhaps a group of Turkish or Bulgarian listeners), and the non-isochronous clips would be drawn from a representative corpus of non-isochronous music from those cultures. If the Turkish/Bulgarian listeners then showed the typical inverse U-shaped relationship between complexity and groove for both isochronous and non-isochronous music, whereas the American/English-speaking participants only showed this trend for isochronous music (and ideally, showed highest groove for low complexity non-isochronous stimuli), that would be compelling evidence for top-down influences of metrical expectations. However, as it is, the present study used a rather unique set of primarily British/American bands and musicians performing non-isochronous meters in metal, math rock, and jazz. The somewhat avant-garde or niche use of

non-isochronous meters by these Western musicians is arguably quite different from the non-isochronous music you would expect to find in Turkey or Bulgaria, which have a rich cultural history of dancing to such meters.

The present manuscript is also unclear about the relative familiarity of non-isochronous vs. isochronous music to the listeners tested. Although familiarity with the specific clips and style is measured and included in the models, there is no description of how familiarity differs between isochronous and non-isochronous clips. My guess is that familiarity ratings for both stimulus sets are not terribly different, given that both stimulus sets were drawn from a sample of Western popular music, which would further undermine the assumption that familiarity ("top-down models of metrical structure") are responsible for the observed effects. Even if the non-isochronous stimuli are slightly less familiar to listeners, the models do not appear to examine the potential relationship between familiarity and the groove x complexity interaction.

Another concern I have is that the range of pulse entropy differs for the two stimulus sets (it is compressed for the non-isochronous meter stimuli), and not much is said about why, nor is this considered in the interpretation of findings. In particular, groove ratings for pulse entropy levels $\sim .6$ are comparable for both stimulus sets, and only appear to differ at slightly higher entropy levels of $.65-.75$. Depending where you draw the line between low vs. moderate complexity, there do not appear to be any non-isochronous stimuli that are low complexity. Thus, it seems we could not observe the inverted U-shaped function for non-isochronous stimuli even if we wanted to. It is interesting that the functions differ for the two stimulus groups, but it is not clear why a positive U-shaped curve would be observed for non-isochronous stimuli. The authors point out that the urge to move is greatest for relatively lower rhythmic complexity among the non-isochronous stimuli, however they say nothing about the finding that the urge to move also increases with complexity for the non-isochronous stimuli. Why would this be the case and how would the authors explain it theoretically?

A final concern I have is that important information is missing from the paper for a variety of details, noted below. A notable concern is that there was minimal information about the musical background questionnaire. Correlations are reported between some of the complexity rating measures and "age stopped [music] training" but there isn't much information about what that means. In particular, it is unclear why age started and age stopped are measured but not total years of music training. What does it mean if a measure is correlated with the age at which an individual stopped taking music lessons?

Minor comments:

1. I suggest avoiding the terms "common" and "uncommon" meters in part because common meter has a different meaning in the context of language/poetry, and also because non-isochronous meters are not actually that uncommon, and this of course depends on where you are in the world! Personally, I prefer terms "isochronous" and "non-isochronous" or "regular" vs. "irregular" or "simple" vs. "complex." Even "4/4" and "non-4/4" would be clearer.

2. The description of pulse entropy is incomplete and only references prior work without describing how the measure works—given that pulse entropy is the central predictor across experiments, the reader should be able to understand (generally) how it is calculated just from reading this paper. Moreover, it is unclear why pulse entropy was chosen as the main indicator of complexity rather than other measures of complexity/syncopation in the literature. Given that subjective complexity measures were obtained in the first experiment, why not instead or also use those to predict groove? The choice of pulse entropy as the main predictor should be clearly motivated and justified.

3. The introduction does not adequately characterize the literature regarding the influence of cultural listening experience and music training on the groove-complexity relationship. It largely ignores studies that have not replicated the U-shaped relationship as well as studies that suggest cultural background/listening experience and musical training have no effect on the preference for moderate-complexity rhythms in groove (Witek, Cameron, Senn and colleagues).

4. The study was apparently run online but relatively little information is provided about how the researchers ensured quality data and compliance. The paper states that participants were instructed to use headphones and that they would not be paid if they simply clicked through the experiment without rating the clip, but no information is provided about how many participants were excluded for this reason or how they were excluded. Was there a headphone check or compliance and attention checks?

5. Although stimuli were selected to be somewhat near a tempo of 120 bpm, there is no information provided about how tempo differed across the different types of stimuli. Casual inspection of the stimulus list suggests (to me) that perhaps the non-isochronous stimuli were slightly slower. Moreover, bpm markings can sometimes be wrong (for instance they might reflect multiples or subdivisions of the felt pulse), and this could influence groove particularly if one stimulus set has lower or higher tempo than the other. Why not also measure event density and examine whether or not it is correlated with groove ratings, or at the very least make sure it is equal across stimulus sets?

Reviewer #2 (Remarks to the Author):

The authors have chosen a statistical method to analyze the data (hierarchical Bayesian regression) that I am not at all familiar with. For this reason, I am neither able to offer a competent critique of the study, nor can I give sound advice on how to improve it. I recommend that editors seek the advice of a reviewer who has the necessary methodological knowledge.

I assume that the vast majority of Communications Psychology readers will not be familiar with the methodology employed in this study. I suggest that authors justify their use of the method, explain why standard methods are not adequate, and offer considerably more detail on the analysis technique, so readers have a chance to follow.

Reviewer #3 (Remarks to the Author):

This paper uses a series of online experiments to explore how pulse clarity and metrical complexity interact when adult listeners are making judgements about how “groovy” a particular song is. In experiment 1, the dissociation between metrical complexity and judgements of rhythmic complexity was validated by listeners rating the stimulus sets on a variety of scales. In experiment 2 and 3, new listeners reported how much each song made them want to move and how much pleasure they experienced when listening. Experiment 2 and 3 uncovered compelling evidence in favour of the predictive coding models for the perception of groove – while moderately syncopated 4/4 music is higher in groove than low or high syncopated music, more complex metered music is most groovy with low levels of syncopation. The explanation here is that 7/8 is hard enough for the Western listener to form predictions about, so the “sweet spot” for groove is achieved even when pulse clarity is high.

This is an elegant study, and the intro was an especially clear and compelling read. The results are informative and interesting, and fill a gap in this (currently very trendy) field. I have a few questions about missing information, and a few suggestions for improved clarity.

Main comments:

- A real strength of this paper is how it provides support for the predictive coding framework. There is nice detail about this in the intro, but I think it's relevance could be highlighted in the discussion
- Were the listeners instructed to move, or refrain from moving, or were they given no explicit instruction?
- Where did the stimuli originally come from? They look like the Janata stimuli but that is unclear in the text. Please also have a secondary rater listen to the music and categorize them as 4/4 vs non 4/4, provide information about inter-rater consistency, and come up with a plan about how to deal with inconsistencies across raters.
- How was “age that music lessons stopped” represented in your model for those who never received training? Was this just missing data? Does that mean that this component of the model was selectively exploring the effects of length of musical training only with participants who did indeed HAVE some musical training (and more training, aka later age at stopping, meant higher predictability/pulse clarity ratings)? I wonder if this can be discussed as more evidence about top-down models. Those with more music training may also have stronger models about predictability, perhaps shifting their “sweet spot”. Can your data address this?
- I'm not sure if I'm convinced that pulse clarity, predictability and perceived complexity measures are capturing something unique. Assuming these are highly correlated, does it make sense to present them as separate DVs or to use these ratings to create a composite score (with alpha reported)?
- Related to these scales, I know that placing methods at the end is how this journal works, but I had to keep flipping back and forth to get the information I needed to make sense of the results. I'd recommend moving some information back up into the results or intro to help the reader avoid this – for example, a bit of information about what the listeners are reporting on would be really useful before trying to interpret the results. Right now it feels like the paper was written intro-methods-results-discussion and then rearranged with little editing.
- How was pulse entropy calculated? There is some mention that some songs were excluded because pulse entropy calculations didn't align well with human-rated pulse clarity, probably because the humans were listening to the bass lines. This is a really important point – should pulse entropy be calculated independently for the low pitch information? Can you filter out the high pitches and recalculate pulse entropy to see if that better aligns with human ratings?
- Although pleasure ratings don't differ as a function of pulse entropy for the complex meter stim, it is interesting that pleasure levels are high. Is this surprising given that these western listeners are probably less familiar with this kind of music?
- I wonder if the non4/4 stim differed only in terms of meter, or if they also differed in another consistent way. Do these stim come from non Western compositional styles? Do the harmonic and melodic sequences feel familiar to a Western listener?
- I would recommend reframing Experiment 2 vs 3 as one experiment with two order conditions. This is functionally what is happening in the analyses, and I don't see any strong theoretical arguments for predicting a difference across random and blocked order conditions to justify presenting them as separate experiments.
- It would be useful to present more background lit surrounding metrical enculturation. Some of the seminal work here is quickly cited, but I think expanding on the developmental trajectory of metrical enculturation would add nicely to your paper. I wonder if the authors might consider going a few steps further in their discussion, as well. Their interpretation for the 7/8 metre conditions is that Western listeners aren't forming strong top-down predictions about these sequences, OR that 4/4 is inherently “easier” to process. I think Experiment 1 argues against this (Western-centric) interpretation of 4/4 being easier, but it is still useful to include these arguments cautiously. So what would be a good way to test this? It seems like having listeners encultured to musical systems that use 7/8 more regularly would fill a really important gap and help the authors with their interpretation. If it is possible to launch this study with such listeners, that would be an amazing addition to the present work. If that is beyond the scope, then fleshing out this future direction more explicitly would be helpful.

Minor comments:

- Please include some mention of the predictive coding model in your abstract.
- The predictive coding framework also helps us form interesting and testable predictions about how groove is perceived as humans become encultured listeners. There are two recent developmental papers exploring groove that I would encourage the authors to read and cite (I'll admit one of these papers is my own, but there is very little out there about how groove

perception develops).

o Cameron, D. J., Caldarone, N., Psaris, M., Carrillo, C., & Trainor, L. J. (2023). The complexity-aesthetics relationship for musical rhythm is more fixed than flexible: Evidence from children and expert dancers. *Developmental Science*, 26(5), e13360.

You do cite this, but with no reference to their developmental findings which are particularly relevant to the arguments made o Kragness, H. E., Anderson, L., Chow, E., Schmuckler, M., & Cirelli, L. K. (2023). Musical groove shapes children's free dancing. *Developmental Science*, 26(1), e13249

- The authors mention the countries that they limited data collection to. Please report the breakdown in % of where your participants lived, as per the listed countries. Did you also collect information about music listening? We are assuming that these listeners have had exposure to Western music, but they may also have extensive exposure to other musical systems.
- There is some note about tempo being manipulated for some stimuli. Please report which stim this happened with.
- Tempo of 10BPM – is that a typo?
- There is a note about participants not getting paid if they didn't provide ratings correctly. Was this a hollow threat? If not, please report the criterion for determining that they did not follow instructions, and report how many participants were excluded for this reason.

Thanks for an interesting read!
Laura Cirelli

EDITORIAL POLICIES

We ask that you ensure your manuscript complies with our editorial policies and reporting requirements.

To that end, we require revised manuscripts to be accompanied by two completed items: a reporting summary that collects information on study design and procedure, and an editorial policy checklist that verifies compliance with all required editorial policies

- <https://www.nature.com/documents/nr-reporting-summary.zip>>Nature Research Reporting Summary
- <https://www.nature.com/documents/nr-editorial-policy-checklist.pdf>>Editorial Policy Checklist

All points on the policy checklist must be addressed. Your revised manuscript can only be sent back to the referees if these checklists are completed and uploaded with the revision.

Notes: If you have submitted a Stage 1 Registered Report, Review, Primer, Comment, or Perspective you do not need to submit these forms. If you have already submitted these forms, you may disregard this request.

** Visit Nature Research's author and referees' website at <http://www.nature.com/authors>>www.nature.com/authors for information about policies, services and author benefits**

If you experience problems in linking your ORCID, please contact the <http://platformsupport.nature.com/>>Platform

Support Helpdesk.

Version 1:

Decision Letter:

Dear Dr Spiech,

Thank you for your patience during the peer-review process. Your manuscript titled "4/4 and More: Groove in Uncommon Meters" has now been seen by 2 reviewers, and I include their comments at the end of this message. We are interested in the possibility of publishing your study in Communications Psychology, but would like to consider your responses to the remaining concerns and assess a revised manuscript before we make a final decision on publication.

We therefore invite you to revise and resubmit your manuscript, along with a point-by-point response to the reviewers. Please highlight all changes in the manuscript text file.

Editorially, we consider it crucial that you interpret your findings in line with your methods. As noted by Reviewer 1, you do not have measurements of expectations or interpretations of top-down rhythm.

I am attaching an Editorial Requests Table that details critical reporting requirements for the revised manuscript. Please attend to each item and ensure your manuscript is fully compliant. If your revised manuscript is not aligned with these requests on major issues, such as those concerning statistics, it may be returned to you for further revisions without re-review.

Please submit the following items:

- Revised manuscript
- Point-by-point response to the referees' comments
- Cover letter (as a separate document)
- [Nature Research Reporting Summary](https://www.nature.com/documents/nr-reporting-summary.pdf)
- Completed Editorial Request Table (attached).

via this link: Link Redacted .

Additional guidance is available in our style and formatting guide [Communications Psychology formatting guide](https://www.nature.com/documents/commpsychol-style-formatting-guide-accept.pdf).

Best regards,

Jennifer Bellingtier

Jennifer Bellingtier, PhD
Senior Editor
Communications Psychology

REVIEWER EXPERTISE:

Reviewer #1 groove, rhythmic complexity, pulse entropy

Reviewer #3 groove, rhythmic complexity, pulse entropy

REVIEWER REPORTS:

Reviewer #1 (Remarks to the Author):

The revised manuscript is an improvement over the first version, for example it now more clearly describes the pulse entropy measure and the assessment of music training. However, the authors chose to rebut most of the more substantive comments and therefore the key problems still remain. The manuscript's narrative is still centered around the claim that it examines "top-down predictions (derived from long-term experience)" even though there is no evidence that top-down predictions vary across individuals for different stimulus sets. The authors simply assert that listeners have more robust mental models of 4/4 than non-4/4 meters and use this to explain the finding that the relationship between groove and complexity/entropy differs for the two types of stimuli. In fact, the authors reveal in the rebuttal letter that familiarity ratings were actually slightly higher for non-4/4 meter than 4/4 excerpts. In my view this completely undermines their story about the role of exposure-based predictions and it suggests other measures of exposure/familiarity are critical if the authors wish to tell the story in this way.

I am also unconvinced by the rebuttal that it is too difficult to find "isolated populations" exposed only to non-4/4 meter— "isolated" populations are not necessary to compare listener groups with relatively more or less exposure to non-4/4 meters. Supplementary analysis 3 does not help with this, as western musicians often show the same biases towards isochronous meters as do Western non-musicians so they are not a good proxy for those with greater exposure to non-4/4 meters. I understand it is challenging to conduct cross-cultural research, but without measures that allow for meaningful comparison of individuals with contrasting life-long listening experiences, I don't see how the authors can make the claims that they do.

To me, the evidence presented in this paper is still interesting because it suggests that when characterizing the relationship between groove and complexity/entropy, not all music produces the inverted U-shaped function-- this has already been shown in prior work but it's still important to document. In this case, the evidence hints at the possibility that measures of complexity (and especially syncopation) are not adequate for predicting groove in all genres of music. However in the current version of the manuscript the authors are telling a much stronger story about exposure-based prediction that is simply not supported by any of the evidence they present.

Reviewer #3 (Remarks to the Author):

Thank you for addressing my comments (and the comments of R1 and R2). I have no other comments and believe the changes have improved the manuscript.

Version 2:

Decision Letter:

Dear Dr Spiech,

Your manuscript titled "4/4 and More: Groove in Uncommon Meters" has now been editorially reviewed, and I am delighted to say that we are happy, in principle, to publish a suitably revised version in Communications Psychology.

We therefore invite you to revise your paper one last time to address the remaining concerns of our reviewers and a list of editorial requests. At the same time we ask that you edit your manuscript to comply with our format requirements and to maximise the accessibility and therefore the impact of your work.

EDITORIAL REQUESTS:

SUBMISSION INFORMATION:

OPEN ACCESS:

*** TRANSPARENT PEER REVIEW:** Communications Psychology uses a transparent peer review system. On author request, confidential information and data can be removed from the published reviewer reports and rebuttal letters prior to publication. If you are concerned about the release of confidential data, please let us know specifically what information you would like to have removed. Please note that we cannot incorporate redactions for any other reasons.

*** CODE AVAILABILITY:** All Communications Psychology manuscripts must include a section titled "Code Availability" at the end of the methods section. We require that the custom analysis code supporting your conclusions is made available in a publicly accessible repository at this stage; please choose a repository that generates a digital object identifier (DOI) for the code; the link to the repository and the DOI must be included in the Code Availability statement. Publication as Supplementary Information will not suffice.

*** DATA AVAILABILITY:**

Link Redacted

Best regards,

Jennifer Bellingtier

Jennifer Bellingtier, PhD
Senior Editor
Communications Psychology

4/4 and More Reviewer Responses

REVIEWER REPORTS:

Reviewer #1 (Remarks to the Author):

The present manuscript reports three experiments examining the relationship between rhythmic complexity, groove, and meter. English-speaking participants were tested on-line and presented with clips of music in 4/4 meter or in non-4/4 meter and rated these clips for rhythmic complexity (Experiment 1), and groove and pleasure (Experiment 2-3). The results replicate existing findings that the relationship between rhythmic complexity and groove in 4/4 music is characterized by an inverse U function with moderate complexity rhythms rated highest in groove and pleasure. By contrast, groove ratings of the non-4/4 musical clips were generally characterized by a positive quadratic function suggesting relatively lower and higher complexity rhythms were rated groovier by listeners. Given the assumption that the listeners were unfamiliar with non-4/4 meters, the authors argue that they provide evidence that a listeners' familiarity with the metrical structure of music influences their experience of groove.

The questions behind this study are fascinating and I love the idea of studying how listeners experience groove in music with non-isochronous meters because. As the researchers point out, this has not yet been studied and it addresses some important theoretical questions about why we experience groove, and the relative importance of listener's cultural experience vs. the structure of the music. Even though I wanted to love this paper, I am not at all convinced that this evidence provides support for the notion that "top-down models of metrical structure shape our embodied responses to rhythms of different complexity as indexed by participants' urge to move ratings" as the authors claim.

Response 1: Thank you! We hope we can convince you to love this paper in the end!

Although the authors somewhat acknowledge this limitation in the discussion, I think it is a serious problem that there is no comparison group with greater exposure to non-isochronous meters. In my view it seems equally if not more likely that the observed differences between ratings of isochronous vs non-isochronous stimuli arise from features of this particular non-isochronous stimulus set and/or perhaps because (maybe?) groove functions differently in non-isochronous than in isochronous music.

Ideally, the design of this study would compare groups of individuals with and without exposure primarily to non-isochronous meters (perhaps a group of Turkish or Bulgarian listeners), and the non-isochronous clips would be drawn from a representative corpus of non-isochronous music from those cultures. If the Turkish/Bulgarian listeners then showed the typical inverse U-shaped

relationship between complexity and groove for both isochronous and non-isochronous music, whereas the American/English-speaking participants only showed this trend for isochronous music (and ideally, showed highest groove for low complexity non-isochronous stimuli), that would be compelling evidence for top-down influences of metrical expectations. However, as it is, the present study used a rather unique set of primarily British/American bands and musicians performing non-isochronous meters in metal, math rock, and jazz. The somewhat avant-garde or niche use of non-isochronous meters by these Western musicians is arguably quite different from the non-isochronous music you would expect to find in Turkey or Bulgaria, which have a rich cultural history of dancing to such meters.

Response 2: We understand the rationale behind the reviewer's suggestion of testing participants with greater exposure to non-4/4 rhythms as a way of further testing the effect of top-down models. However, this approach is more difficult than it might appear intuitively. First, because of the globalization of media and the internet, most people have considerable exposure to Western music. This is a common problem for cross-cultural studies, and why they often go to considerable lengths to test isolated populations (c.f., Jacoby et al., 2024). Second, while non-4/4 meters may be more common in the music of particular regions or cultures, these meters are not necessarily the most common metrical form, and not all of the people from that region would be familiar with this music. To do this kind of a study properly would require developing very specific stimulus sets and recruiting and testing very carefully selected groups of participants who we could confirm have greater experience with the non-4/4 music chosen.

In our case, the choice of primarily Western genres was a conscious one meant to ensure that our Western participants would have roughly similar exposure to the musical properties of the selected excerpts with the primary differences relating to the rhythmic complexity and the meter.

Another approach to addressing this question of exposure to non-4/4 meters is to test trained musicians who would be likely to have greater experience with a broader range of music, potentially including music with a wider variety of metric structures. Therefore we split our sample from Experiments 2 and 3 into those with more than 10 years of music playing ($N = 49$) and those with 0 years of playing ($N = 72$). The results between the two groups did not differ significantly (see plots below) which would explain why musician-related covariates did not improve model fits or interact significantly with our variables of interest. This has now been included in Supplementary Analysis 3. Future work addressing this issue could examine jazz musicians who are more specifically trained in complex meters to further confirm the contribution of top-down effects.

We have now commented on the potential usefulness of future work including cross-cultural studies or samples of musicians with specific training to the Discussion (**page 18**).

The present manuscript is also unclear about the relative familiarity of non-isochronous vs. isochronous music to the listeners tested. Although familiarity with the specific clips and style is measured and included in the models, there is no description of how familiarity differs between isochronous and non-isochronous clips. My guess is that familiarity ratings for both stimulus sets are not terribly different, given that both stimulus sets were drawn from a sample of Western popular music, which would further undermine the assumption that familiarity (“top-down models of metrical structure”) are responsible for the observed effects. Even if the

non-isochronous stimuli are slightly less familiar to listeners, the models do not appear to examine the potential relationship between familiarity and the groove x complexity interaction.

Response 3: We did not ask listeners to rate overall familiarity with 4/4 compared with non-4/4 meters because it is not obvious how to ask non-musicians to rate familiarity with different metrical structures. Creating such a measure would require a separate study with considerable development and validation. This is something we could consider for future work.

As the reviewer noted, we did ask listeners to rate each musical clip for song familiarity and style preference which were meant to control for their documented effects on groove. However, in response to the reviewer's question, we compared 4/4 and non-4/4 ratings pooled across all three experiments in order to maximize sensitivity. Paired sample t-tests comparing the familiarity and style preference ratings between 4/4 and non-4/4 stimuli and found no effect of meter on style preference ratings ($t(382) = 0.5879, p = 0.5569$) and only a small effect on familiarity ratings ($t(382) = 4.7618, p < 0.001, d = 0.24$) where clips in non-4/4 meters were rated as slightly *more familiar* than songs in 4/4.

Another concern I have is that the range of pulse entropy differs for the two stimulus sets (it is compressed for the non-isochronous meter stimuli), and not much is said about why, nor is this considered in the interpretation of findings. In particular, groove ratings for pulse entropy levels $\sim .6$ are comparable for both stimulus sets, and only appear to differ at slightly higher entropy levels of $.65-.75$. Depending where you draw the line between low vs. moderate complexity, there do not appear to be any non-isochronous stimuli that are low complexity. Thus, it seems we could not observe the inverted U-shaped function for non-isochronous stimuli even if we wanted to. It is interesting that the functions differ for the two stimulus groups, but it is not clear why a positive U-shaped curve would be observed for non-isochronous stimuli. The authors point out that the urge to move is greatest for relatively lower rhythmic complexity among the non-isochronous stimuli, however they say nothing about the finding that the urge to move also increases with complexity for the non-isochronous stimuli. Why would this be the case and how would the authors explain it theoretically?

Response 4: This is an important question. However, as now reported in the Methods (page 22) average pulse entropy value doesn't differ between 4/4 and non-4/4 stimuli ($t(48.49) = 0.85605, p = 0.3962; BF = 0.366$). Further, to assess the possible effect of differences in the range of the pulse entropy values, we removed three low entropy excerpts and one high-entropy excerpt from the 4/4 stimulus set to produce an equivalent range to the non-4/4 set. When we re-run the urge to move and pleasure analyses with this truncated 4/4 set, the best models still support an interaction between pulse entropy and meter (Urge to Move: $BF = 6.24 \times 10^6$; Pleasure: $BF = 3.08 \times 10^5$). For Urge to Move, this interaction was driven by the quadratic term for pulse entropy where, for the non-4/4 tracks, the same positive quadratic relationship was observed (β_1

= 207.61, 95% CI [169.58, 247.30], BF = 1.57 x 10⁸) while there was only a negative linear term for the 4/4 tracks ($\beta_1 = -142.31$, 95% CI [-185.37, -100.69], BF = 956.39). For pleasure, the interaction was driven by the linear term of pulse entropy where, for the non-4/4 tracks, the same null effect was observed (BF = 0.608 for the linear term, BF = 0.042 for the quadratic term) and only a negative linear relationship was observed for the 4/4 tracks ($\beta_1 = -145.37$, 95% CI [-180.56, -109.07], BF = 1.96 x 10⁵). In summary, truncating the range of pulse entropy only seems to trim off the decrease in groove for lower complexity stimuli in 4/4 while maintaining elevated groove for mid-complexity stimuli. This additional analysis is now noted on page 15 of the text and is included in the supplementary materials.

Regarding the slight uptick in urge to move ratings at the higher end of rhythmic complexity for non-4/4 tracks, this is puzzling to us as well and doesn't fit neatly within current theories. That said, we binned the pulse entropy of the non-4/4 tracks into categorical tertiles of low, mid, and high rhythmic complexity and compared whether this slight uptick in the urge to move was statistically significant. While high rhythmic complexity tracks were rated significantly higher than mid rhythmic complexity tracks ($t(239) = 2.625$, $p = 0.009222$, $d = 0.17$), the magnitude of the effect is very small (the difference in means is ~1.876 points a continuous scale from 0 to 100). For this reason, we're hesitant to strongly interpret this finding without further replication. Perhaps it's possible that without a metric model, participants interpret "how much the music made me want to move" as more arousal-based at high rhythmic complexity rather than as more beat-based movements, but this would require replication with a different questionnaire.

A final concern I have is that important information is missing from the paper for a variety of details, noted below. A notable concern is that there was minimal information about the musical background questionnaire. Correlations are reported between some of the complexity rating measures and “age stopped [music] training” but there isn't much information about what that means. In particular, it is unclear why age started and age stopped are measured but not total years of music training. What does it mean if a measure is correlated with the age at which an individual stopped taking music lessons?

Response 5: We understand that the “age stopped training” is confusing. This was an issue with participant responses to our questionnaire – some participants reported the age that they stopped playing music without reporting the year that they started playing which made it impossible to estimate how many years they had been actively playing music. Thus, we thought it was safer to simply report the data that we could confidently assess, like the age they started formal training (if they had any), the age they reported stopping their music playing (if they did indeed stop playing), the amount of hours per week spent playing music, and the composite Barcelona Musical Reward Questionnaire score.

However, while the age that music playing stopped seemed to significantly correlate with dependent variables in Experiment 1, this correlation was not strong enough to improve model fit and the Supplementary Analysis 3 on explicit training did not reveal any striking differences in the ratings of Experiments 2 and 3 either. This is now clarified on page 20.

Minor comments:

1. I suggest avoiding the terms “common” and “uncommon” meters in part because common meter has a different meaning in the context of language/poetry, and also because non-isochronous meters are not actually that uncommon, and this of course depends on where you are in the world! Personally, I prefer terms “isochronous” and “non-isochronous” or “regular” vs. “irregular” or “simple” vs. “complex.” Even “4/4” and “non-4/4” would be clearer.

Response 6: We agree that common/uncommon may not have been the best descriptors. However, not all of the meters fit the other categories, so we have opted for “4/4 and non-4/4” instead.

2. The description of pulse entropy is incomplete and only references prior work without describing how the measure works—given that pulse entropy is the central predictor across experiments, the reader should be able to understand (generally) how it is calculated just from reading this paper. Moreover, it is unclear why pulse entropy was chosen as the main indicator of complexity rather than other measures of complexity/syncopation in the literature. Given that subjective complexity measures were obtained in the first experiment, why not instead or also

use those to predict groove? The choice of pulse entropy as the main predictor should be clearly motivated and justified.

Response 7: We have now included a more detailed description of how pulse entropy is calculated and why we chose this metric on **page 5 and page 21** of the manuscript. The full rationale is included below.

Music information retrieval metrics like pulse entropy provide a simple, data-driven way to extract acoustic features directly from an audio file. We chose pulse entropy because calculating other measures of syncopation or complexity requires transcriptions of all instrumental lines in a given excerpt. Transcription is problematic, not only because it would be prohibitively time-consuming, but also because it would be prone to error, as stereo mixes of commercial recordings are notoriously difficult to parse into individual streams (e.g. via stem separation tools) without some (unknown) degree of signal distortion and information loss. Furthermore, transcriptions are subject to interpretation because they can be ambiguous and subject to musicological interpretation. Second, to our knowledge, syncopation score indices have primarily been used as complexity measures in studies of simplified solo drumkit patterns - where almost always the drumkit has been typically further reduced to just the hi-hat, snare, and kick drums in order to decrease the complexity of the calculations - such as in Witek et al, (2014; 2015) or Matthews et al. (2019; 2022). As far as we can tell, no studies yet have applied such syncopation calculations to large polyphonic ensemble contexts featuring multiple instruments, for it becomes a far more complex, and correspondingly more subject to interpretation, when deciding what the total syncopation index of all instruments in an ensemble should be. This is because syncopation indices identify and count the syncopations for each instrumental line. Therefore it would be necessary to assign an arbitrary weight to each syncopation in each line and then sum them all together. The problem with this is that we find no clear justification for adding up the individual syncopations within and across all patterns beyond an intuitive understanding that more syncopations equal more complexity. This, however, is problematic. Take for example, the case of a pattern where all events are syncopated, such as a phase-shifted stream of quarter note guitar strokes found in ska music; although not likely to be considered perceptually complex by listeners, such a pattern would produce a relatively high syncopation score using typical syncopation indices.

Pulse entropy measures the estimated predictability of the beat in a given piece. Specifically, the function from the MIR Toolbox we used detects event onsets from the audio file. Then it runs a sliding window autocorrelation on these onsets to construct a curve to determine how self-similar the timing between events is over time, and then calculates the entropy of the peaks of this curve. We opted for pulse entropy specifically because it measures the entropy of the rhythmic onsets in a piece of music and we were interested in how groove relates to this objective, acoustic measure of rhythmic complexity rather than to subjective appraisals. Pulse entropy metrics are also

typically highly related to listener ratings of perceived rhythmic complexity (Singer, Jacoby, Hendler, & Granot, 2023; Kantan, Alecu, & Dahl, 2021; Toussaint & Trochidis, 2018), which is the construct we aim to explore. As shown in Figure 1, pulse entropy values in the current study are strongly related to listener ratings of perceived complexity, predictability and pulse clarity, indicating that this metric is clearly related to listener perceptions. Further, when the urge to move and pleasure models in Experiment 3 are re-run using the average perceived rhythmic complexity values from Experiment 1, the results do not fundamentally change.

3. The introduction does not adequately characterize the literature regarding the influence of cultural listening experience and music training on the groove-complexity relationship. It largely ignores studies that have not replicated the U-shaped relationship as well as studies that suggest cultural background/listening experience and musical training have no effect on the preference for moderate-complexity rhythms in groove (Witek, Cameron, Senn and colleagues).

Response 8: Thank you, we have now included more references reviewing the effects of listening experience and musical training on groove in the introduction (page 3).

4. The study was apparently run online but relatively little information is provided about how the researchers ensured quality data and compliance. The paper states that participants were

instructed to use headphones and that they would not be paid if they simply clicked through the experiment without rating the clip, but no information is provided about how many participants were excluded for this reason or how they were excluded. Was there a headphone check or compliance and attention checks?

Response 9: Participants appeared to understand and follow the instructions and no one was excluded for non-compliance. To verify compliance, rating data were reviewed offline to identify anyone whose ratings did not span more than two and a half standard deviations of their average. No one was excluded on this basis. There was no headphone check.

5. Although stimuli were selected to be somewhat near a tempo of 120 bpm, there is no information provided about how tempo differed across the different types of stimuli. Casual inspection of the stimulus list suggests (to me) that perhaps the non-isochronous stimuli were slightly slower. Moreover, bpm markings can sometimes be wrong (for instance they might reflect multiples or subdivisions of the felt pulse), and this could influence groove particularly if one stimulus set has lower or higher tempo than the other. Why not also measure event density and examine whether or not it is correlated with groove ratings, or at the very least make sure it is equal across stimulus sets?

Response 10: Similar to the pulse entropy, the average tempo did not statistically differ between the two meter conditions ($t(53.162) = 0.067, p = 0.947$). To control for the subjectivity of bpm markings, we calculated the event density of each clip with the MIR Toolbox and tested whether it differed between the two meter conditions. Welch's two sample t -tests revealed that average event density did not statistically differ between the two meter conditions ($t(50.963) = 1.1201, p = 0.2679$). Thus, it seems unlikely that our results were confounded by tempo or event density. This is now noted in Supplementary Analysis 2.

Reviewer #2 (Remarks to the Author):

The authors have chosen a statistical method to analyze the data (hierarchical Bayesian regression) that I am not at all familiar with. For this reason, I am neither able to offer a competent critique of the study, nor can I give sound advice on how to improve it. I recommend that editors seek the advice of a reviewer who has the necessary methodological knowledge.

I assume that the vast majority of Communications Psychology readers will not be familiar with the methodology employed in this study. I suggest that authors justify their use of the method, explain why standard methods are not adequate, and offer considerably more detail on the analysis technique, so readers have a chance to follow.

Response 1: On **pages 6-7**, we note that Bayesian statistics are necessary to quantify the evidence *for* the null hypothesis; frequentist statistics can only quantify the evidence for the alternative hypothesis. This was crucial to determine whether there was *no* interaction between Meter and Pulse Entropy in Experiment 1 and *no* effect of covertly presenting the stimuli in blocks of 4/4 and non-4/4 meters in Experiment 3. We have added more detail about the methodology to **pages 6-7 and 24-25** as well as suggested reading for those unfamiliar with Bayesian statistics.

Reviewer #3 (Remarks to the Author):

This paper uses a series of online experiments to explore how pulse clarity and metrical complexity interact when adult listeners are making judgements about how “groovy” a particular song is. In experiment 1, the dissociation between metrical complexity and judgements of rhythmic complexity was validated by listeners rating the stimulus sets on a variety of scales. In experiment 2 and 3, new listeners reported how much each song made them want to move and how much pleasure they experienced when listening. Experiment 2 and 3 uncovered compelling evidence in favour of the predictive coding models for the perception of groove – while moderately syncopated 4/4 music is higher in groove than low or high syncopated music, more complex metered music is most groovy with low levels of syncopation. The explanation here is that 7/8 is hard enough for the Western listener to form predictions about, so the “sweet spot” for groove is achieved even when pulse clarity is high.

This is an elegant study, and the intro was an especially clear and compelling read. The results are informative and interesting, and fill a gap in this (currently very trendy) field. I have a few questions about missing information, and a few suggestions for improved clarity.

Response 1: Thank you so much!

Main comments:

- A real strength of this paper is how it provides support for the predictive coding framework. There is nice detail about this in the intro, but I think it’s relevance could be highlighted in the discussion

Response 2: Thank you, we have added more details about predictive coding to **page 16** in the Discussion section.

- Were the listeners instructed to move, or refrain from moving, or were they given no explicit instruction?

Response 3: We know from previous work that tapping along with the beat can result in higher groove ratings (Spiech et al, 2022; <https://osf.io/preprints/psyarxiv/fw7mh>). Therefore, in order to avoid suggesting the idea of moving to the participants and potentially influencing their ratings, and to keep the experiment as naturalistic as possible, participants were given no explicit instruction.

- Where did the stimuli originally come from? They look like the Janata stimuli but that is unclear in the text. Please also have a secondary rater listen to the music and categorize them as 4/4 vs non 4/4, provide information about inter-rater consistency, and come up with a plan about how to deal with inconsistencies across raters.

Response 4: Authors C.S. and G.S.C. compiled the stimuli and conferred to reach consensus on the categorization of the tracks as 4/4 vs non-4/4. In response to the reviewer's question, we asked Dr. Anne Danielsen (<https://www.uio.no/ritmo/english/people/management/anneda/>), an expert in the analysis of rhythm and meter to confirm these categorizations. Overall inter-rater reliability was very high (Cronbach's $\alpha = 0.959$ [0.867, 1.000]), with only two discrepant tracks. Re-analysis either omitting or recategorizing these two tracks did not change the pattern of results.

- How was "age that music lessons stopped" represented in your model for those who never received training? Was this just missing data? Does that mean that this component of the model was selectively exploring the effects of length of musical training only with participants who did indeed HAVE some musical training (and more training, aka later age at stopping, meant higher predictability/pulse clarity ratings)? I wonder if this can be discussed as more evidence about top-down models. Those with more music training may also have stronger models about predictability, perhaps shifting their "sweet spot". Can your data address this?

Response 5: As noted in Response 5 to Reviewer 1, this was an issue with participant responses to our questionnaire. Some participants reported the age that they stopped playing music without reporting the year that they started playing. Thus, we thought it was safer to simply report the data that we could confidently assess, like the age they started formal training (if they had any), the age they reported stopping their music playing (if they did indeed stop playing), the amount of hours per week spent playing music, and the composite Barcelona Musical Reward Questionnaire score. What it seems to track is a combination of age and active musicianship (Age Stopped was set to the participant's current age if they reported that they still play music). That said, while the age that music playing stopped seemed to significantly correlate with dependent variables in Experiment 1, this correlation was not strong enough to improve model fit or interact with our Pulse Entropy or Meter manipulations and the Supplementary Analysis 3 on explicit training did not reveal any striking differences in the ratings of Experiments 2 and 3

either. For these reasons, we're hesitant to interpret this as strong evidence of a training or expertise effect. This is now clarified on page 20.

- I'm not sure if I'm convinced that pulse clarity, predictability and perceived complexity measures are capturing something unique. Assuming these are highly correlated, does it make sense to present them as separate DVs or to use these ratings to create a composite score (with alpha reported)?

Response 6: We agree that pulse clarity, predictability and perceived complexity are largely capturing the same quality (rhythmic complexity) and indeed these measures are highly related (Cronbach's alpha = 0.827 [0.819, 0.835]). We presented them separately in Exp 1 in order to support the robustness of the pulse entropy metric (i.e., that it wasn't dependent on idiosyncrasies related to wording of the different ratings). We nevertheless have added Cronbach's alpha to page 9 of the results section to emphasize that we did intend for them to capture the same phenomenon.

- Related to these scales, I know that placing methods at the end is how this journal works, but I had to keep flipping back and forth to get the information I needed to make sense of the results. I'd recommend moving some information back up into the results or intro to help the reader avoid this – for example, a bit of information about what the listeners are reporting on would be really useful before trying to interpret the results. Right now it feels like the paper was written intro-methods-results-discussion and then rearranged with little editing.

Response 7: Thank you for this helpful suggestion, we hope that it reads more clearly now. Specifically, we moved more methodological details like how pulse entropy is calculated and the interpretation of certain statistics to the introduction section on pages 5-7.

- How was pulse entropy calculated? There is some mention that some songs were excluded because pulse entropy calculations didn't align well with human-rated pulse clarity, probably because the humans were listening to the bass lines. This is a really important point – should pulse entropy be calculated independently for the low pitch information? Can you filter out the high pitches and recalculate pulse entropy to see if that better aligns with human ratings?

Response 8: We calculated pulse entropy using the Music Information Retrieval Toolbox in Matlab. Put simply, the algorithm reads in the audio file, detects the event onsets, runs a sliding autocorrelation window on them to construct a curve to determine how self-similar the timing between events is over time, and then calculates the entropy of the peaks of this curve.

As described in the Methods (pages 21-22), three stimuli were excluded from all analyses because of incongruencies between pulse entropy measures and human ratings.

To address the reviewer's question, we low-pass filtered all of the stimuli using the "mirfilterbank(..., '2Channels')" function, which (from 1.8.1. manual): "performs a computational simplification of the filterbank using just two channels, one for low-frequencies, below 1000 Hz, and one for high-frequencies, over 1000 Hz (Tolonen and Karjalainen, 2000)". We then recalculated pulse entropy again, and it doesn't seem to map onto participants' ratings any differently:

- Although pleasure ratings don't differ as a function of pulse entropy for the complex meter stim, it is interesting that pleasure levels are high. Is this surprising given that these western listeners are probably less familiar with this kind of music?

Response 9: We agree that it is interesting that pleasure levels are relatively high. While perhaps unexpected, we think that this is a positive feature of the stimuli because it means that listeners do not appear to find them to be globally strange or unpleasant.

- I wonder if the non4/4 stim differed only in terms of meter, or if they also differed in another consistent way. Do these stim come from non Western compositional styles? Do the harmonic and melodic sequences feel familiar to a Western listener?

Response 10: The stimuli are primarily composed in Western styles drawn from pop, jazz, and rock genres. We ensured that the musical genres were balanced between meter conditions so that any differences in harmonic and melodic progressions would be minimized. Additionally, we included participants' ratings of song familiarity and style preference ratings as random effects.

- I would recommend reframing Experiment 2 vs 3 as one experiment with two order conditions. This is functionally what is happening in the analyses, and I don't see any strong theoretical arguments for predicting a difference across random and blocked order conditions to justify presenting them as separate experiments.

Response 11: The motivation for Experiment 3 was to rule out implicit priming effects between 4/4 and non-4/4 trials. Therefore we feel that keeping the experiments separate better explains our logic while designing the experiment. This motivation has now been better explicated on page 6.

- It would be useful to present more background lit surrounding metrical enculturation. Some of the seminal work here is quickly cited, but I think expanding on the developmental trajectory of metrical enculturation would add nicely to your paper. I wonder if the authors might consider going a few steps further in their discussion, as well. Their interpretation for the 7/8 metre conditions is that Western listeners aren't forming strong top-down predictions about these sequences, OR that 4/4 is inherently "easier" to process. I think Experiment 1 argues against this (Western-centric) interpretation of 4/4 being easier, but it is still useful to include these arguments cautiously. So what would be a good way to test this? It seems like having listeners encultured to musical systems that use 7/8 more regularly would fill a really important gap and help the authors with their interpretation. If it is possible to launch this study with such listeners, that would be an amazing addition to the present work. If that is beyond the scope, then fleshing out this future direction more explicitly would be helpful.

Response 12: As discussed above in Response 2 to Reviewer 1, testing participants with greater exposure to non-4/4 rhythms is more difficult than it might intuitively appear. First, Western music is ubiquitous in most parts of the world, hence the heroic efforts that cross-cultural studies go to find isolated populations. Second, while non-4/4 rhythms may be more common in certain cultures, they are not necessarily the norm, nor is music (often traditional) written in these meters what most people from these cultures commonly listen to. As described in the same response, we did test trained musicians who are likely to have greater exposure to non-4/4 meters, and their results did not differ from our unselected sample (See Supplementary Analysis 3). Future studies could examine musicians with jazz or other specific types of training to address this question. We have now included a brief statement of these ideas in the Discussion (page 18).

Minor comments:

- Please include some mention of the predictive coding model in your abstract.

Response 13: We have now included the concept of predictions in the abstract.

- The predictive coding framework also helps us form interesting and testable predictions about how groove is perceived as humans become encultured listeners. There are two recent developmental papers exploring groove that I would encourage the authors to read and cite (I'll admit one of these papers is my own, but there is very little out there about how groove perception develops).

o Cameron, D. J., Caldarone, N., Psaris, M., Carrillo, C., & Trainor, L. J. (2023). The complexity-aesthetics relationship for musical rhythm is more fixed than flexible: Evidence from children and expert dancers. *Developmental Science*, 26(5), e13360.

You do cite this, but with no reference to their developmental findings which are particularly relevant to the arguments made

o Kragness, H. E., Anderson, L., Chow, E., Schmuckler, M., & Cirelli, L. K. (2023). Musical groove shapes children's free dancing. *Developmental Science*, 26(1), e13249

Response 14: Thank you for these helpful suggestions, we have included them on pages 18-19 of the discussion now.

- The authors mention the countries that they limited data collection to. Please report the breakdown in % of where your participants lived, as per the listed countries. Did you also collect information about music listening? We are assuming that these listeners have had exposure to Western music, but they may also have extensive exposure to other musical systems.

Response 15: The plot below is now included in the Supplementary Materials. We did not explicitly ask about exposure to other musical systems but, as reported in Response 3 to Reviewer 1, rated style preference (how often do you listen to this style of music) did not differ between the 4/4 and non-4/4 clips ($t(382) = 0.5879, p = 0.5569$). While we cannot rule out that they had more exposure to other musical systems, we can at least say that our Western participants had similar self-rated exposure to Western music composed in 4/4 as Western music composed in non-4/4 meters.

- There is some note about tempo being manipulated for some stimuli. Please report which stim this happened with.

Response 16: This is now noted with asterisks in Supplementary Table 1.

- Tempo of 10BPM – is that a typo?

Response 17: This has been corrected to 110 bpm.

- There is a note about participants not getting paid if they didn't provide ratings correctly. Was this a hollow threat? If not, please report the criterion for determining that they did not follow instructions, and report how many participants were excluded for this reason.

Response 18: It is standard practice for online studies on Prolific to inform participants that if they do not complete the study as instructed the researcher can withhold payment. In this case, we told participants to try to use the full range of the rating scales. As a compliance check we tested whether participants' ratings spanned more than two and a half standard deviations of their average. No participants were excluded based on this criterion.

Thanks for an interesting read!
 Laura Cirelli

Response 19: Thanks Laura!

Response to Reviewers

Reviewer #1 (Remarks to the Author):

The revised manuscript is an improvement over the first version, for example it now more clearly describes the pulse entropy measure and the assessment of music training. However, the authors chose to rebut most of the more substantive comments and therefore the key problems still remain. The manuscript's narrative is still centered around the claim that it examines "top-down predictions (derived from long-term experience)" even though there is no evidence that top-down predictions vary across individuals for different stimulus sets. The authors simply assert that listeners have more robust mental models of 4/4 than non-4/4 meters and use this to explain the finding that the relationship between groove and complexity/entropy differs for the two types of stimuli.

I am also unconvinced by the rebuttal that it is too difficult to find "isolated populations" exposed only to non-4/4 meter— "isolated" populations are not necessary to compare listener groups with relatively more or less exposure to non-4/4 meters. Supplementary analysis 3 does not help with this, as western musicians often show the same biases towards isochronous meters as do Western non-musicians so they are not a good proxy for those with greater exposure to non-4/4 meters. I understand it is challenging to conduct cross-cultural research, but without measures that allow for meaningful comparison of individuals with contrasting life-long listening experiences, I don't see how the authors can make the claims that they do.

In fact, the authors reveal in the rebuttal letter that familiarity ratings were actually slightly higher for non-4/4 meter than 4/4 excerpts. In my view this completely undermines their story about the role of exposure-based predictions and it suggests other measures of exposure/familiarity are critical if the authors wish to tell the story in this way.

To me, the evidence presented in this paper is still interesting because it suggests that when characterizing the relationship between groove and complexity/entropy, not all music produces the inverted U-shaped function-- this has already been shown in prior work but it's still important to document. In this case, the evidence hints at the possibility that measures of complexity (and especially syncopation) are not adequate for predicting groove in all genres of music. However in the current version of the manuscript the authors are telling a much stronger story about exposure-based prediction that is simply not supported by any of the evidence they present.

Response:

We understand the point of the reviewer that we have made strong claims that our findings support the role of top-down models on groove perception. We have now moderated these claims in the manuscript and provided additional evidence that Western listeners have demonstrably stronger internal models of 4/4 meters compared to non-4/4 meters, thus supporting our interpretation.

We have also specifically described how future studies suggested by the reviewer could test our interpretations and explained why these studies would be necessary to confirm our results.

However, we also feel strongly that the current evidence supports our interpretation and that the suggested future studies, particularly cross-cultural comparisons, cannot be reasonably done in the context of the current set of experiments.

As described in the Introduction to our paper, this set of experiments was designed as a first step in testing the possible influence of top-down predictions derived from long-term musical experience on the perception of musical groove. To our knowledge, no previous study has examined this question. Thus, our interpretation of the current data generates new hypotheses that need to be tested further in future studies. Without our findings that Western listeners do not exhibit the same pattern of groove response for non-4/4 meters, we could not then raise the question of whether other listeners with greater exposure to these meters might show the canonical response.

Below we respond to specific concerns raised by the reviewer in more detail.

Question 1: Evidence for stronger models for 4/4 meter in Western listeners

As reviewed in the Introduction and Discussion of this paper, the premise that the Western listeners tested in this study have a stronger model of 4/4 compared to non-4/4 meters related to their musical experience is based on substantial evidence showing that 4/4 is the most common meter in Western music (London, 2012) and that in fact 4/4 and related meters are common world-wide (Jacoby et al., 2024; Savage et al., 2015). More specifically, 4/4 meters are dominant in modern-era Western pop music, particularly the type of music that has typically been used in previous studies of musical groove (Stewart, 2000). Based on this evidence, we think that it is reasonable to conclude that the Western listeners in our studies have greater experience, and thus likely a stronger internal model for 4/4 meters allowing us to interpret our findings in this light. This logic is similar to developmental studies examining the impact of musical background on pitch perception (Hannon & Trainor, 2007) or native language exposure on phoneme perception (Romberg and Saffran, 2010; Polka and Werker, 1994; Werker and Hensch, 2014). In fact, it would be very difficult to make the converse argument that our listeners do *not* have greater experience with 4/4 meters. Further, the assumption that listeners from different musical environments develop differing musical expectations based on exposure is the same one that underlies the reviewer's request that we conduct a cross-cultural study looking at listeners from musical backgrounds that would presumably promote stronger internal models for non-4/4 meters.

We have now added more evidence of greater exposure to 4/4 than non-4/4 meters in Westerners to the Introduction on pages 4 and 5.

Despite this evidence, we agree with the reviewer that our current interpretation of our findings should be tested in future studies that either relate individual differences in the strength of the top-down metrical model to the experience of groove, or which compare groups with greater long-term exposure to non-4/4 meters who would be hypothesized to possess stronger models for certain non-4/4 meters.

We have now further revised the Discussion to state even more clearly that this type of future study would be required to test our ideas and have outlined how these studies could be designed. More detailed description of such studies is given below.

Question 2: Follow-up studies

First, we agree that it could be very interesting to examine the relationship between perceived groove for 4/4 vs non-4/4 excerpts and the degree of familiarity with these meters and thus the strength of individual participants' top-down metrical models. A number of groove studies have asked people to rate how familiar they are with a particular piece of music or the style of the piece (Senn, 2018; 2021; 2023; Witek et al., 2014; Janata et al., 2012), but none have asked listeners to rate familiarity with meter (the high-level temporal structure of a piece). Non-musicians can of course readily judge whether a piece of music or genre is familiar or not – but they may not be able to identify the specific aspects that make it familiar: is it the meter, melodic structure, harmony, instruments used, or something else? Therefore, we would need to develop a way of testing metrical familiarity that is not confounded with these other factors. As we stated in the previous response to this same question, there is no currently available measure for directly assessing familiarity with the meter of a piece of music. Creating such a metric would be very useful for future studies and we are already considering how best to do it, however this will take considerable time and testing.

Second, we also agree that conducting a cross-cultural comparison of people with greater long-term experience with non-4/4 meters would be a strong test of our hypothesis that top-down predictions based on experience influence groove perception. However, as we described in our previous response to the same question, we do not believe that a rigorous and interpretable cross-cultural study can be done without finding appropriate collaborators and developing appropriate materials.

Conducting an appropriate cross-cultural study would not mean simply asking a group of listeners from a purportedly different musical background to rate our current stimuli. The stimuli used in these experiments were specifically chosen to be drawn from the Western popular music repertoire because we wanted to test Western listeners who would be broadly familiar with the musical styles. To conduct an appropriate cross-cultural study, we would need to create a similar stimulus set drawn from a different musical tradition. This would require working closely with musicians or musicologists familiar with this tradition to source appropriate musical clips, and perhaps translate the rating scales.

Further, we would need to recruit and test participants who we could confirm have greater experience with the non-4/4 music chosen. In the same way that the reviewer emphasizes that we may not be able to assume that all Western listeners are more exposed to 4/4, we cannot assume that all people from a particular geographic region are all more exposed to non-4/4 meters, first because most people have considerable exposure to Western music and second because although non-4/4 meters may be more common in the music of particular region or culture, these meters are not necessarily the most common metrical form, and not all of the people from that region would be familiar with this music.

We are currently seeking collaborators to work with us on such a study, but it will take considerable testing and development.

Question 3: Influence of familiarity

The reviewer is concerned that the song familiarity ratings differed between the 4/4 and non-4/4 excerpts such that non-4/4 pieces were rated as somewhat more familiar than 4/4 excerpts. First, it is important to reiterate that song familiarity and style preference ratings were included in all models as random effects to control for their potential influence on the complexity and groove ratings.

Because the reviewer had previously asked about familiarity, we separately compared song familiarity ratings between 4/4 and non-4/4 stimuli across all three experiments and found that, on average, participants rated the non-4/4 excerpts as somewhat more familiar: approximately 1.97 points higher out of 100. As we stated in our previous response, musical style preference *did not differ* between 4/4 and non-4/4 musical clips (mean difference = 0.229; Bayesian paired-sample t -test: $BF_{10} = 0.0681$, $BF_{01} = 14.685$; frequentist paired-sample t -test: $t(382) = 0.5879$, $p = 0.5569$).

Question 4: Analysis of data from trained musicians

In a supplementary analysis we tested a sample of trained musicians who would be expected to have greater exposure to non-4/4 meters through training. We found that trained musicians showed the same pattern of results for non-4/4 excerpts as non-musicians, suggesting that our findings may generalize to those with greater experience with less common meters.

The reviewer questioned this idea, stating that “as western musicians often show the same biases towards isochronous meters as do Western non-musicians so they are not a good proxy for those with greater exposure to non-4/4 meters.” While this is broadly true, this assertion is not entirely consistent with current evidence showing that like non-musicians Western-trained musicians show strong peaks for simple meters related to 4/4, but they also display greater peaks for non-4/4 meters (e.g., 7/8; Yates, 2017; Jacoby & McDermott, 2017). See panel E of Figure 7 extracted from Jacoby and McDermott (2017) below.

Thus, while not as strong a piece of evidence as could be provided by a well-designed cross-cultural study, we believe that this analysis is still informative. In fact, these results support an alternative interpretation of our findings in suggesting that there may be a broader bias in favor of 4/4 meters that is independent of experience. This alternative is already included in the Discussion (pages 18-19).

We have now added additional information supporting the rationale for this analysis in the Supplementary Materials and strengthened the presentation in the Discussion.